# Positivity and Health Locus of Control: Key Variables to Intervene on Well-Being of Cardiovascular Disease Patients

**DOI:** 10.3390/jpm13050873

**Published:** 2023-05-22

**Authors:** Bárbara Luque, Naima Z. Farhane-Medina, Marta Villalba, Rosario Castillo-Mayén, Esther Cuadrado, Carmen Tabernero

**Affiliations:** 1Maimonides Biomedical Research Institute of Cordoba (IMIBIC), 14004 Córdoba, Spain; bluque@uco.es (B.L.); z62famen@uco.es (N.Z.F.-M.); esther.cuadrado@uco.es (E.C.); carmen.tabernero@usal.es (C.T.); 2Department of Psychology, University of Cordoba, 14071 Cordoba, Spain; 3Institute of Neurosciences of Castilla y León (INCYL), University of Salamanca, 37007 Salamanca, Spain; 4Department of Social Psychology and Anthropology, University of Salamanca, 37005 Salamanca, Spain

**Keywords:** cardiovascular disease (CVD), health locus of control, positivity, psychological well-being, health-related quality of life

## Abstract

Psychological well-being is a good predictor of several health outcomes in cardiovascular disease patients (adherence, quality of life, and healthy behaviors). The perception of health control and a positive orientation seem to have a beneficial effect on health and well-being. Therefore, the aim of this study was to investigate the role of the health locus of control and positivity in the psychological well-being and quality of life of cardiovascular patients. A total of 593 cardiac outpatients completed the Multidimensional Health Locus of Control Scale, the Positivity Scale and the Hospital Anxiety and Depression Scale at baseline (January 2017) and 9 m later (follow-up; *n* = 323). A Spearman rank correlation coefficient and a structural equation modeling approach were determined to explore the relationships between those variables both cross-sectionally and longitudinally. A cross-sectional correlation analysis at baseline revealed that the internal health locus of control and positivity were negatively associated with anxiety (*r*_s_ = −0.15 and −0.44, *p*s < 0.01) and depression (*r*_s_ = −0.22 and −0.55, *p*s < 0.01) and positively associated with health-related quality of life (*r*_s_ = 0.16 and 0.46, *p*s < 0.01). Similar outcomes were found at follow-up and in longitudinal correlations. According to the path analysis, positivity was found to be negatively associated with anxiety and depression level at baseline (β = −0.42 and −0.45, *p*s < 0.001). Longitudinally, positivity was negatively associated with depression (β = 0.15, *p* < 0.01) and together with the internal health locus of control, was positively associated with health-related quality of life (β = 0.16 and 0.10, respectively, *p*s < 0.05). These findings suggest that focusing on the health locus of control and especially positivity may be crucial in enhancing the psychological well-being of patients in the context of cardiac care. The potential impact of these results on future interventions is discussed.

## 1. Introduction

Cardiovascular disease (CVD) is a broad term that encompasses various medical conditions affecting the heart and blood vessels [1]. These conditions include coronary artery disease, heart failure, arrhythmias, valvular heart disease, and peripheral artery disease, among others [1]. CVD is highly prevalent worldwide, and this trend appears to be increasing. Recent data from the Global Burden of Cardiovascular Diseases and Risk Factors [2] show that the total prevalence of CVD has nearly doubled in the last three decades, from 271 million in 1990 to 523 million in 2019. The increase in prevalence has been accompanied by a significant rise in disability-adjusted life years and years lived with disability from 17.1 million to 34.4 in the same period, as well as an alarming mortality rate among affected patients [2]. With its different typologies, CVD remains the leading cause of death worldwide [3]. To such an extent that according to the World Health Organization [4], in 2019, 17.9 million people died due to CVD, representing 32% of global deaths. The enormous burden of this health condition [5] has led to the need for working at a preventive level, from health promotion to the implementation of interventions that reduce associated risk factors [6]. This aspect has become a real challenge for healthcare systems from all disciplines that address these types of diseases [7,8].

There are several risk factors related to CVD, but psychosocial factors are particularly relevant due to their influence in this disease [6,7,9]. According to the European Society of Cardiology [10], some of the psychosocial risk factors noted are low socio-economic status, lack of social support, Type D personality, stress at work and in family life, hostility, depression, and anxiety [9]. Therefore, from a biopsychosocial approach of health, interventions for CVD patients should also address their psychological well-being to mitigate the emotional consequences of the diagnosis [6,8,11]. To design effective integrative interventions for CVD patients, it is crucial to explore which psychological variables influence cardiovascular health (CVH) and well-being, as well as the role of personality dispositions in these variables. This approach could be beneficial both in the prevention and treatment of CVD, considering individual differences in psychological profiles [12,13]. For these reasons, incorporating a psychosocial approach in cardiac rehabilitation could improve the clinical management of CVD and have a positive impact on patient outcomes [14].

### 1.1. CVD and Psychological Well-Being and Quality of Life

Psychological well-being is a complex term to define, but it is typically associated with positive thoughts and emotions that individuals experience regarding their life satisfaction and overall sense of worth [15]. Therefore, it refers to optimal psychological functioning, which includes a combination of emotional aspects (e.g., happiness and experiencing positive emotions) as well as higher-level functions such as resilience, coping, and emotional regulation [15,16]. In contrast, psychological distress is composed of constructs such as anxiety, depression, anguish, or hostility [17].

Recent research has shown the association and relevance of psychological well-being or distress among patients with CVD [11,18,19]. Studies have found that increased levels of well-being were related to improved CVH [11] and lower odds of mortality due to a cardiac event [6,20]. Conversely, psychological distress, specifically depression and anxiety, has been bidirectionally linked to CVD [9,21]. Meta-analyses and longitudinal studies have provided evidence that depression and anxiety are risk factors for CVD, with an association between them and an increased risk of developing heart diseases such as ischemic heart disease or coronary heart disease [22,23,24]. In addition, the diagnosis of CVD may exacerbate anxiety and depressive symptoms in these patients. Some studies have reported a high prevalence of depression among patients with coronary artery disease, with 34% of them experiencing moderate to severe depression, which could negatively affect their prognosis [25]. A meta-analysis performed by Gathright et al. [26] found that depression was a predictor of all-cause mortality in heart failure. Furthermore, patients with coronary heart disease or heart failure and depressive symptoms are more likely to have a lower quality of life and a greater risk for recurrent cardiovascular events and mortality [23]. At the same time, studies addressing psychological distress and CVD have showed some sex differences. Women with CVD reported greater and more severe symptomatology of anxiety and depression than men with CVD [27,28], highlighting the need to consider these differences when designing and applying treatments for CVD patients.

CVD can have a significant impact on patients’ lives, not only causing psychological distress mentioned above but also physical symptoms such as reduced mobility, pain, and fatigue [29]. These symptoms may become chronic and can negatively affect their quality of life [13]. Previous studies have shown the association between a CVD diagnosis and a lower HRQoL [30,31], which includes physical, mental, and social factors, as well as subjective perceptions of health and well-being [32]. At the same time, lower HRQoL is associated with other CVD risk factors such as a reduced adherence to medication and an in-creased risk of recurrent cardiovascular events [33,34,35]. These findings have led to a wide body of research studying the role of HRQoL as an important variable in the context of chronic conditions, such as CVD, in order to better understand and intervene on its impact.

The impact of psychological distress on the quality of life of individuals with cardiovascular disease emphasizes the need to intervene in their emotional and psychological care [6,7,8,9,14]. Further research is necessary to identify key variables that promote their psychological well-being and provide protective effects against the disease.

### 1.2. CVD and Personality Dispositions: Positivity and Health Locus of Control

Personality dispositions are consistent patterns of thoughts, feelings, and behaviors that characterizes an individual and are relatively stable over time and across different contexts [36]. Some personality dispositions have been found to be related to well-being and quality of life, such as positivity and the health locus of control (HLC). Positivity is a psychological construct that provides insights into individuals’ overall sense of wellness as it can be defined by factors such as self-esteem, satisfaction with life, and optimism [37]—factors that have been studied by their association with chronic conditions. For instance, self-esteem has been found to mediate the relationship between life satisfaction and lower depression in CVD patients [38]. Optimism, on the other hand, seems to favor other psychological and cognitive mechanisms that promote better cardiovascular health (CVH) [11,18,39], reduce the risk of cardiovascular events [40], and facilitate the engaging of health-related behaviors [39]. In addition, several studies have found that positivity is a significant predictor of psychological variables related to psychological well-being and quality of life, such as depression and anxiety [11,37,41], as well as to the prognosis of the disease, reducing the rate of rehospitalization and mortality [41]. Therefore, encouraging individuals to adopt a more positive outlook on life, including their illness, would be necessary to help them feel more in control and self-efficacious in coping with their situation [18,40]. Thus, another much related and equally important variable would be the health locus of control.

Improving a CVH prognosis involves making lifestyle changes [42], which can be influenced by various factors, including the personality disposition of HLC [43]. HLC refers to people’s beliefs about their ability to control their health, which can be either internal or external [43]. Internal HLC refers to individuals’ perception of having control over their health, while external HLC refers to the belief that external factors, such as genetics, chance, or other people including family and physicians, have control over one’s health [44]. According to the Health Locus of Control Theory, individuals’ health-related behaviors are associated with their perception of their ability to overcome health problems. Internal HLC has been found to be linked to engage in positive and protective health behaviors [45,46]. On the other hand, individuals with external HLC may have a lower sense of personal responsibility for one’s own health, which may result in worse health outcomes and a poorer disease prognosis [47].

The literature shows the significant role of psychological variables and personality traits in better understanding and conceptualizing the onset and consequences of CVD in a person’s life. However, there is still a need to further investigate the specific mechanisms that underlie the association between these variables. Such research could provide valuable information for designing future psychological treatments and enhancing the effectiveness of cardiac rehabilitation programs, leading to improved psychological well-being and quality of life for CVD patients.

### 1.3. Aim and Hypotheses

The aim of this research was to explore the influence of HLC and positivity variables on psychological well-being, considering levels of anxiety and depression, and HRQoL in patients with CVD over time. For this purpose, the evaluations of these variables were carried out at two different times, thus being able to obtain results of the same variables in a first phase (baseline), and after approximately 9 months in a second phase (follow-up). The hypotheses proposed for this study were as follows (Figure 1):

**Hypothesis** **H1.**
*Sociodemographic variables are significantly associated with the dispositional variables, psychological well-being and the HRQoL.*


**Hypothesis** **H2.**
*Patients with higher levels of positivity will have lower levels of anxiety and depression, and therefore greater psychological well-being as well as higher health-related quality of life, both cross-sectionally (H2a) and longitudinally (H2b).*


**Hypothesis** **H3.**
*The patients with higher levels of internal HLC will have lower levels of anxiety and depression, and therefore greater psychological well-being and HRQoL, both cross-sectionally (H3a) and longitudinally (H3b).*


**Hypothesis** **H4.**
*The internal HLC and positivity, given the stability of personality dispositions, will maintain similar correlations with psychological well-being and HRQoL both at baseline and at follow-up.*


**Hypothesis** **H5.**
*Positivity will predict the levels of psychological well-being and HRQoL variables (dependent variables) in patients, both cross-sectionally (H5a) and longitudinally (H5b).*


**Hypothesis** **H6.**
*The internal HLC will predict the levels of psychological well-being and HRQoL variables (dependent variables) in patients, both cross-sectionally (H6a) and longitudinally (H6b).*


## 2. Materials and Methods

### 2.1. Participants and Procedure

This study included 593 CVD patients (*M* = 64.75, *SD* = 9.07) recruited from the Cardiology Unit of the University Reina Sofía Hospital (Córdoba, Spain) who participated in the CORDIOPREV study [48,49] using a convenience sampling method. The inclusion and exclusion criteria of this study followed those of the primary study. Patients that had already suffered a first cardiac event and were diagnosed with an established coronary heart disease (e.g., unstable coronary disease, acute myocardial infarction, chronic coronary disease, and unstable angina) were included. CVD patients that had experienced a clinical event in the last 6 months were excluded. The study sample characteristics had been previously published [48,49].

A longitudinal study was designed to test the hypotheses proposed. A battery of questionnaires was administered to the participants to assess HLC, positivity, psychological well-being through the components of anxiety and depression, and HRQoL at two different times. The baseline assessment started in January 2017 and the follow-up evaluation was conducted approximately 9 months later. At baseline, the sample was composed of 593 patients, and at follow-up of 323 CVD patients. Sociodemographic characteristics measured are shown in Table 1.

The current study was approved by the corresponding Research Ethics Committees (June 2015). Prior to their participation, all patients were informed about this study’s objectives and assured that their involvement would be voluntary and anonymous. Those who consented to participate provided written informed consent. Participants completed the questionnaires using tablets in a designated room at the hospital. The surveys were conducted using the Unipark program (v. 10.9), which is an online survey software available through the Questback academic program. Sociodemographic characteristics measured are shown in Table 1.

### 2.2. Instruments

Sociodemographic ad hoc questionnaire. The study participants were asked to provide sociodemographic details such as their sex, age, employment status, marital status (whether they had a partner), educational background, and economic status.

Multidimensional Health Locus of Control Scale (MHLC-S [46]). The MHLC-S evaluates the locus of control for health. It is composed of four factors according to whom the control is assigned, which in this study are termed: internal HLC; chance HLC; doctors HLC; and other people HLC. Although the original scale contains 24 items (e.g., “*I am directly responsible for my condition getting better or worse*”), in this study, a short version of the scale composed of 12 items was used (each factor contains 3 items). It uses a Likert response format of 7 points in a range of 1 (strongly disagree) to 7 (strongly agree). The original study shows a measure with Cronbach’s alpha between 0.66 and 0.77. Due to the low reliability of the doctor’s factor in this study, it was excluded from the analysis. However, after deleting this subscale, Confirmatory Factor Analyses indicated a good model fit of the scale at both measurement moments. The fit indices at T1 were: χ2 (24) = 84.86, *p* < 0.001; RMSEA (90% CI) = 0.07 (0.05, 0.08); CFI = 0.96; TLI = 0.95; GFI = 0.97; and AGFI = 0.94. At T2, the fit indices were χ2(21) = 49.59, *p* = 0.002; RMSEA (90% CI) = 0.06 (0.04, 0.08); CFI = 0.97; TLI = 0.96; GFI = 0.97; and AGFI = 0.93.

*Positivity Scale* (P-scale [37]). This scale evaluates the personal tendency to interpret life and vital experiences from a positive point of view. It is a unidimensional scale that contains eight items (e.g., “*I have great faith in the future*”) ranging from 1 (strongly disagree) to 5 (strongly agree). A higher score indicates greater positivity. Previous studies have reported adequate internal consistency of the measure in different countries, including Spain, with a Cronbach’s alpha of 0.81 [37].

*Hospital Anxiety and Depression Scale* (HADS [50], Spanish validation from Terol et al. [51]). The HADS is a self-administered scale that allows the evaluation of psychological well-being considering the total score on the scale, as well as from the two factors that compose it: anxiety (HADS-A; e.g., “*I get sort of a frightened feeling as if something awful is about to happen*”) and depression (HADS-D; e.g., *“I feel as if I am slowed down”*). This measure is composed of 14 items distributed in anxiety and depression subscales, each one having 7 items and with a 7-point frequency scale ranging from 1 (*never*) to 7 (*every day*). A higher score indicates higher levels of anxiety and depression, respectively, and therefore a lower level of psychological well-being when considering the total score of the scale. Previous studies have reported adequate internal consistency: α = 0.78 in HADS-A, α = 0.82 in HADS-D, and α = 0.89 in the general scale [51].

The Short Form-12 Health Survey (SF-12 [52], Spanish validation from Failde et al. [53]). The SF-12 is a self-report questionnaire that assesses health-related quality of life. It consists of 12 items that measure 8 domains related to health: physical functioning, role limitations due to physical problems, bodily pain, general health perceptions, vitality, social functioning, role limitations due to emotional problems, and mental health. These domains are subdivided in a mental health component (MCS; six items: e.g., “*have you had any problems with your work or other regular daily activities as a result of your emotional problem (such as feeling depressed or anxious)?*”) and a physical health component (PCS; six items: e.g., “*Does your health now limit you in walking several blocks?*”). From the twelve items, eight were presented on a 5-point Likert-type scale and four in a dichotomous format that required a ‘yes’ or ‘no’ response. Higher scores indicate better health-related quality of life. Previous studies have reported adequate internal consistency: α = 0.85 in PCS and α = 0.78 in MCS [54].

### 2.3. Statistical Analysis

Descriptive statistics were conducted to know the frequencies of the sociodemographic characteristics. Subsequently, a Kolmogorov–Smirnov test was conducted to determine whether the data were normally distributed. The results conclude that the assumption of normality was violated in all evaluated scales. Therefore, we performed Spearman’s rank correlation analyses to measure the association between the variables cross-sectionally and longitudinally. Then, we conducted a path analysis using the structural equation modelling (SEM) approach to further explore the explanatory capacity of the HLC and positivity on psychological well-being, anxiety and depression factors, and health-related quality of life. The model’s adequacy was evaluated by means of several metrics, including the chi-squared statistic (χ2), the comparative fit index (CFI), the Tucker–Lewis index (TLI), the root mean square error of approximation (RMSEA), and the standardized root mean square residual (SRMR). For model evaluation, we followed the recommendations of Schermelleh-Engel et al. [55]. According to these authors, an acceptable model fit is indicated by a χ2/df value that is equal to or less than 3, as well as CFI and NNFI values that are equal to or greater than 0.95, and RMSEA values that are less than 0.08, accompanied by a confidence interval (CI) that is in close proximity to RMSEA. The independent variables were “health locus of control (HLC)” and “positivity”, while the dependent variables were “psychological well-being” assessed through levels of “anxiety” and “depression”, and HRQoL. A descriptive analysis and Spearman’s correlations were performed using the statistic software SPSS (v.28) and to estimate the path coefficients, we used the software package AMOS (v.13). To interpret correlation results, we considered Cohen’s (1988) [56] suggestions, where a correlation coefficient of 0.1 to 0.3 was considered small, 0.3 to 0.5 was moderate, and greater than 0.5 was large. For both analyses, we set the significance level at *p* < 0.05.

## 3. Results

### 3.1. Sociodemographic Characteristics of Participants

Sociodemographic features of the participants are presented in Table 1. The study sample was majorly composed of men [85.7%]. Regarding labor status, most of the participants were retired [67.1%], followed by having a full-time job [20.1%], houseworkers [5.1%], unemployed [4.9%], and part-time workers [2.9%]. Regarding marital status, most patients had a partner [88.9%]. Finally, the highest number of respondents had received middle educational qualification [56%], followed by high educational level [38.1%], low [3%], very high [2.7%], and very low [0.2%].

### 3.2. Cross-Sectional Analysis

In the first correlation analysis, data were obtained on the associations between all variables (at baseline and at follow-up independently). The significant correlations between the study variables were as follows (Table 2): at baseline, the other people HLC correlated negatively with depression (*r*_s_ = −0.22) as well as with psychological distress (*r*_s_ = −0.17); the internal HLC correlated negatively with anxiety (*r*_s_ = −0.15) and depression (*r*_s_ = −0.22) and with psychological distress (*r*_s_ = −0.20) and positively with all the HRQoL (global, *r*_s_ = 0.16; MCS, *r*_s_ = 0.14; PCS, *r*_s_ = 0.15). Moreover, the internal HLC was positively associated with the other people HLC factor (*r*_s_ = 0.26) and positivity (*r*_s_ = 0.13); the chance HLC correlated negatively with depression only (*r*_s_ = −0.10). Finally, positivity correlated positively with the internal and other people HLC (*r*_s_ = 0.13, *r*_s_ = 0.20, respectively) and all HRQoL factors (global, *r*_s_ = 0.46; MCS, *r*_s_ = 0.51; PCS, *r*_s_ = 0.34), and negatively with psychological distress (*r*_s_ = −0.55), depression (*r*_s_ = −0.55), and anxiety (*r*_s_ = −0.44). At follow-up, the only differences were that the internal HLC did not positively correlate with the PCS HRQoL. Other people HLC correlated negatively with anxiety (*r*_s_ = −0.19) and positively with MCS HRQoL (*r*_s_ = 0.16). Thus, most of the correlation results remained stable over time.

### 3.3. Longitudinal Analysis

The second correlation analysis was conducted to identify any statistically significant relationships between the evaluated variables at baseline and at follow-up (Table 3). Regarding the independent variables, the three HLC factors at baseline significantly correlated with themselves at follow-up (internal HLC, *r*_s_ = 0.44; other people HLC, *r*_s_ = 0.47; and chance HLC, *r*_s_ = 0.43). Only other people HLC was significantly associated with positivity at follow-up (*r*_s_ = 0.18). In contrast, baseline positivity correlated with the variables’ positivity (*r*_s_ = 0.54), other people HLC (*r*_s_ = 0.23), and internal HLC (*r*_s_ = 0.17) at follow-up. On the other hand, with respect to the association between independent and dependent variables, several significant data were found. Baseline scores of the other people HLC correlated negatively with follow-up psychological distress (*r*_s_ = −0.17), anxiety (*r*_s_ = −0.13), depression (*r*_s_ = −0.22), and PCS HRQoL (*r*_s_ = −0.14); the internal HLC showed similar outcomes, but with the difference that this factor correlated positively with the global (*r*_s_ = 0.13) and MCS HRQoL (*r*_s_ = 0.16) and not with the PCS HRQoL and anxiety; the chance HLC did not correlate significantly with any dependent variable. Finally, baseline positivity correlated negatively with follow-up anxiety (*r*_s_ = −0.35), depression (*r*_s_ = −0.36), and psychological distress (*r*_s_ = −0.39) and positively with all HRQoL factors (global, *r*_s_ = 0.33; MCS, *r*_s_ = 0.36; PCS, *r*_s_ = 0.26).

After performing correlation analyses, it was observed that the diagonal, i.e., the variables (dependent and independent) at baseline, correlated positively with themselves with a mostly large magnitude [56] at follow-up. Additionally, associations were found between the independent and dependent variables in these correlations (Table 3). Based on these findings, further comprehensive evaluations were conducted to examine the predictive potential of positivity and HLC variables on psychological well-being, anxiety, depression, and HRQoL both cross-sectionally and longitudinally. To achieve this, a path analysis was performed. The model (Figure 1) demonstrated a strong fit to the data, with the following indices: χ2 (37, *n* = 323) = 41.636, *p* = 0.276; CMIN/DF = 1.125; CFI = 0.995; TLI = 0.991; AGFI = 0.957, GFI = 0.980, RMSEA = 0.020, 95% CI [0.001, 0.046]). Figure 2 displays the standardized parameter estimates.

As it can be seen in Figure 2, at baseline, positivity was negatively related to anxiety (β = −0.42, *p* < 0.001) and depression (β = −0.45, *p* < 0.001). The internal HLC was positively related to the other people HLC (β = 0.24, *p* < 0.001), and negatively with chance HLC (β = −0.20, *p* < 0.001) but not with the dependent variables. Other people HLC was negatively related to depression (β = −0.12, *p* < 0.01). At follow-up, no significant relationship was found between the independent (positivity and HLC) and dependent variables (anxiety, depression, and HRQoL). Longitudinally, positivity negatively predicted anxiety (β = −0.15, *p* < 0.01) and positively the global HRQoL (β = 0.16, *p* < 0.01). The internal and other people HLC also predicted the global HRQoL positively (β = 0.10, *p* < 0.05) and negatively (β = −0.16, *p* < 0.001), respectively.

Besides these associations, other relationships between dependent variables both cross-sectionally and longitudinally were also found. For instance, anxiety and depression at baseline positively predicted anxiety (β = 0.48, *p* < 0.001) and depression (β = 0.39, *p* < 0.001) at follow-up, respectively. Cross-sectionally, at follow-up, anxiety was positively related to depression (β = 0.40, *p* < 0.001) and both anxiety and depression were negatively related to the global HRQoL (β = −0.32 and −0.25, respectively, both *p*s < 0.001).

Finally, the results of the path analysis showed some significant associations between the sociodemographic variables age, sex, and educational level that need to be acknowledged. Being female was associated with higher levels of anxiety (β = 0.21, *p* < 0.001) and lower levels of internal HLC (β = −0.13, *p* < 0.05). Age was positively associated with other people (β = 0.19, *p* < 0.001) and chance HLC (β = 0.15, *p* < 0.01), and negatively with anxiety both at baseline (β = −0.22, *p* < 0.001) and at follow-up (β = −0.14, *p* < 0.01). Finally, the educational level was positively related to positivity at baseline (β = 0.26, *p* < 0.001) and to lower levels of depression at follow-up (β = −0.10, *p* < 0.01).

## 4. Discussion

The aim of this study was to analyze the influence of HLC and positivity on the general psychological well-being, based on the levels of anxiety and depression, and HRQoL of patients with CVD. The results found were in line with what was hypothesized in H2a and H3a. Both positivity and internal HLC correlated significantly and negatively with the dependent variables, anxiety and depression, and positively with psychological well-being and HRQoL, both at baseline and at follow-up. With respect to H2b, the results were also as expected since positivity still showed the same significant relationships with the dependent variables after 9 months (longitudinal correlation). The study’s findings were also in line with H3b because internal HLC correlated positively with psychological well-being and HRQoL factors (except PCS) and negatively with depression; however, there was no significant relationship with anxiety. Furthermore, it should be noted that the factor of other people HLC showed similar associations to the internal HLC regarding the dependent variables, proving to be more related to the well-being of patients with CVD than anticipated. Finally, the results of the correlation analysis supported H4. The personality dispositions of internal HLC and positivity maintained similar correlations with anxiety, depression, and HRQoL at both assessment points, indicating consistent and stable associations between the analyzed variables.

With respect to H5a and H6a, the results partially support the hypotheses. Specifically, positivity predicted psychological well-being at baseline, but not at follow-up. Additionally, no significant results were found between internal HLC and the other dependent variables, neither at baseline nor at follow-up. As expected, the relationship between the independent variables, positivity in this case, with the dependent variables was negative. This means that higher levels of positivity corresponded to lower levels of anxiety, depression, and psychological discomfort, ultimately leading to greater well-being. These findings align with previous research, underscoring the significance of positivity in the psychological well-being of CVD patients [11,18,19]. Maintaining a positive outlook can help prevent emotional states that may become pathological when prolonged, such as elevated levels of anxiety and depression. Improved psychological well-being may, in turn, lead to a better prognosis for CVD patients. Results such as those obtained in the meta-analysis of DuBois et al. [41] have shown the association of positive emotions with the reduction of mortality in CVD, which is another sign that these emotions benefit psychological well-being and therefore a cardiovascular prognosis. Alessandri et al. [57] stated in their work that positivity acted as a variable that promotes positive affect (this being a component of subjective well-being) and serves as a buffer for depression and negative affect. The introduction of the present study mentioned the importance of optimism, which is one of the components of positivity. Some studies, such as Sahoo et al. [58], highlighted the protective role of personality traits such as optimism against the development of CVD; similarly, others argued that a higher level of optimism led to a lower risk of CVD mortality and lower levels of anxiety and depression [59].

Regarding H5b, the results of this study partially support this hypothesis, which suggests that positivity has the potential to predict lower levels of anxiety and higher levels of HRQoL over time. The findings provide further evidence of the significance of positivity, not only for its influence on emotional aspects at specific moments but also for its long-term impact, in line with previous studies [60]. These results indicate that positivity may be a useful tool for improving patient outcomes and highlight the need for further research on this construct as an intervention for CVD patients, given its potential to positively impact their long-term health and well-being.

Finally, H6b is also partially supported by the findings of this study. However, considering the importance of the internal HLC manifested in the literature, it was anticipated that this variable would be a strong predictor across all dependent variables over time. Nevertheless, the capacity of prediction was only found related to HRQoL. Although different from what was expected, these results demonstrate a significant approach to the subject. Firstly, this study’s results align with previous research that outlines the association between internal HLC and HRQoL in CVD patients [47,61]. The influence of internal HLC on the quality of life of CVD patients has also been demonstrated by its impact on modifiable risk factors. Internal HLC has been associated with increased physical activity dedication, decreased alcohol consumption, and even lower mortality rates among cardiac patients [61,62]. Furthermore, internal HLC has been linked to various health-related outcomes associated to HRQoL in chronic patients. For instance, it has been shown to promote better maintenance of physical function after hospitalization [63], increase resilience, reduce stress, enhance physical activity and lower drug consumption among patients with pain conditions [64], improve self-efficacy levels of patients with heart failure [65], and have a positive effect on some diabetes-related cardiovascular risk factors (e.g., glycemic control) [66], among others. These findings, along with our results, suggest that internal HLC has a significant impact on factors related to HRQoL that are relevant to CVD patients. This highlights the need to prioritize psychological treatments that promote and enhance this construct, which may prove effective in improving HRQoL and reducing cardiovascular risk factors.

On the other hand, a negative prediction of other people HLC was found over HRQoL. These findings are consistent with previous studies [61,67]. Blaming external factors for one’s health may lead to a perception of a lack of control and autonomy in the health–disease process. In the context of cardiac care, this may negatively impact CVD patients’ self-efficacy, leading to poor adherence to medication and other essential healthy behaviors, such as a healthy diet and physical activity, which are crucial for effective disease management and a good quality of life [68,69,70,71]. These results support the need to consider the influence of external HLC when implementing policies and interventions aimed at promoting healthy behaviors of these patients [72]. Therefore, the findings of this study underscore the importance of promoting patient empowerment, autonomy, and patient responsibility in healthcare interventions in order to mitigate the lack of perceived control over their own health that may affect the quality of life of these patients.

### 4.1. Practical Implications

This longitudinal study has allowed us to test the predictive hypotheses proposed supporting the potential relevance of a clinical intervention based on providing patients with a positive approach and strengthening their internal locus of control. In line with previous literature that has reported a well-known relationship between psychological variables and CVD risk factors and CVH [6,9], the present findings provide evidence of the role of these psychological variables in the quality of life and anxiety and depressive symptoms in chronic patients [73,74]. Positivity and HLC may also influence the management of CVD, given their association with other psychological variables that are key to a better understanding and management of the disease [11,18,39], such as self-efficacy [65]. This association may, hence, promote CVH (e.g., healthier diet, better adherence to treatment, and quitting smoking) with significant positive outcomes among these patients [68]. Therefore, healthcare providers may incorporate strategies that focus on building positivity and self-perception of control as part of a comprehensive treatment plan for CVD patients to improve their overall health and well-being. Accordingly, the results of this study add interesting information to be considered in future CVD interventions such as cardiac rehabilitation programs, as well as emerging healthcare trends, such as tailored interventions [75] or telemedicine [76].

### 4.2. Limitations and Future Research

There are some limitations in this study that need to be acknowledged and considered for future research. Firstly, the use of self-reporting questionnaires, even with validated instruments and guaranteed confidentiality and anonymity, can introduce bias into the data by relying solely on subjective reporting. Future studies may incorporate a multi-method assessment in order to obtain more accurate information and reduce social desirability when collecting data (e.g., including external validation, honesty scales, etc.). In addition, the magnitude of the reported correlations could be considered as weak and/or moderate according to Cohen (1988) [56]. However, recent literature criticizes the use of Cohen’s cut-off and proposes a more flexible classification [77], which would give our results greater power and validity. In any case, given the potential relevance of these results for therapeutic interventions, future studies with larger samples would be needed to detect stronger relationships between these variables. In line with this, to obtain a better understanding of the trajectory of outcomes and increase the statistical power, future studies may benefit from including more follow-up evaluations. The underrepresentation of women in this study sample may hinder the generalizability of the findings to the female population with CVD. Additionally, the possible influence of other variables has not been studied (for example, a stressful event between assessments). Given the demonstrated relevance of anxiety, depression, and HRQoL in chronic and CVD patients, it is utterly important to conduct further research in this area to explore and study the variables that impact patients’ psychological well-being and to measure the disease prognosis associated with psychological states. This would enable the development of new psychological interventions that consider the influence of these variables, aiming to enhance the well-being of these patients and improve their quality of life.

## 5. Conclusions

This study highlights the important role that positivity and HLC play in psychological health outcomes for CVD patients. These findings suggest that promoting a positive orientation and internal HLC may lead to improved psychological well-being, reduced anxiety and depression levels, and enhanced HRQoL among these patients. In conclusion, this study’s results underline the importance of considering patients’ psychological well-being in the context of cardiac rehabilitation, and suggest that interventions focused on a psychological approach may be beneficial for enhancing CVH and a better prognosis for these patients. Further studies are required in this direction in order to empirically investigate the effectiveness of incorporating this approach in cardiac care.

## Figures and Tables

**Figure 1 jpm-13-00873-f001:**
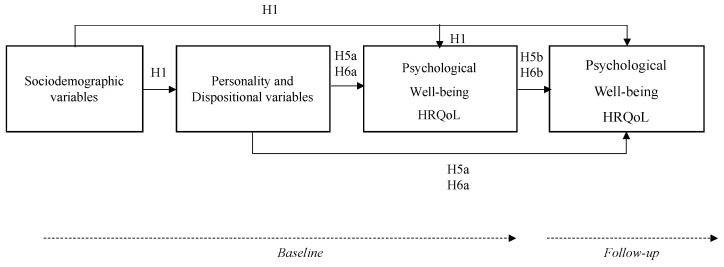
Proposed research hypothesis model. Sociodemographic variables: age, sex, economic and educational level, etc. Personality and dispositional variables: positivity and health locus of control. Psychological well-being: levels of anxiety and depression; HRQoL: health-related quality of life. The model does not include H2, H3, and H4 as they refer to correlations between variables rather than predictive relationships.

**Figure 2 jpm-13-00873-f002:**
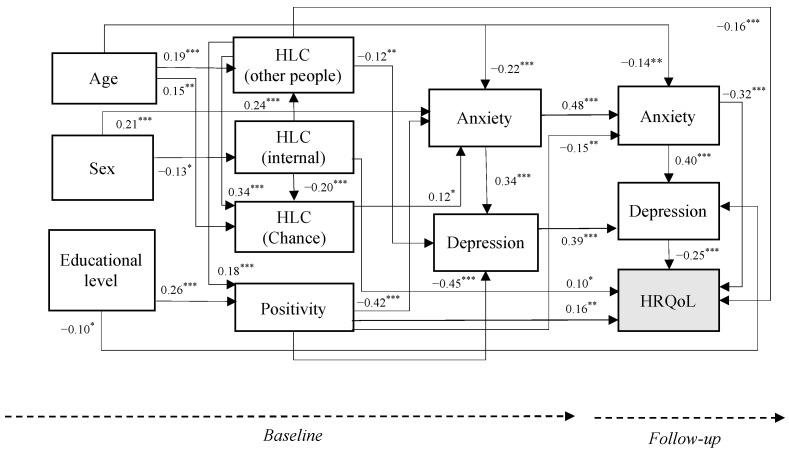
Standardized model parameter estimates (* *p* < 0.05; ** *p* < 0.01; *** *p* < 0.001). HLC = health locus of control; HRQoL = health-related quality of life. The figure only shows significant paths (*n* = 323).

**Table 1 jpm-13-00873-t001:** Patients’ sociodemographic characteristics (*n* = 593).

Sociodemographic Characteristics	Frequencies	(%)
Sex		
Male	508	85.7
Female	85	14.3
Age, Mean (SD) = 64.75 (9.07)		
Employment status		
Unemployed	29	4.9
Part-time worker	17	2.9
Full-time worker	119	20.1
Retired	398	67.1
Housework	30	5.1
Partner		
Yes/With	527	88.9
No/Without	66	11.1
Educational level		
Very low	1	0.2
Low	18	3.0
Middle	332	56.0
High	226	38.1
Very high	16	2.7
Economic level		
Very low	8	1.3
Low	90	15.2
Middle	460	77.6
High	35	5.9

**Table 2 jpm-13-00873-t002:** Cross-sectional correlation analysis of all variables at baseline (a) and at follow-up (b).

(a)	Baseline (*n* = 593)									
	1	2	3	4	5	6	7	8	9	10
1. Other people HLC	1									
2. Internal HLC	0.26 **	1								
3. Chance HLC	0.30 **	−0.02	1							
4. Positivity	0.20 **	0.13 **	0.07	1						
5. HADS	−0.17 **	−0.20 **	−0.05	−0.55 **	1					
6. Anxiety	−0.09	−0.15 **	−0.02	−0.44 **	0.91 **	1				
7. Depression	−0.22 **	−0.22 **	−0.10 *	−0.55 **	0.86 **	0.58 **	1			
8. SF-12	−0.02	0.16 **	−0.02	0.46 **	−0.63 **	−0.54 **	−0.59 **	1		
9. SF-12 (MCS)	0.05	0.14 **	0.01	0.51 **	−0.73 **	−0.65 **	−0.65 **	−0.88 *	1	
10. SF-12 (PCS)	−0.03	0.15 **	−0.06	0.34 **	−0.43 **	−0.38 **	−0.44 **	−0.91 **	0.61 **	1
**(b)**	**Follow-up (*n* = 323)**									
	1	2	3	4	5	6	7	8	9	10
1. Other people HLC	1									
2. Internal HLC	0.25 **	1								
3. Chance HLC	0.39 **	0.05	1							
4. Positivity	0.29 **	0.15 **	0.10	1						
5. HADS	−0.29 **	−0.25 **	−0.11	−0.48 **	1					
6. Anxiety	−0.19 *	−0.21 *	−0.06	−0.43 **	0.93 **	1				
7. Depression	−0.36 **	−0.27 **	−0.19 **	−0.47 **	0.86 **	0.64 **	1			
8. SF-12	0.10	0.15 **	−0.01	0.39 **	−0.56 **	−0.53 **	−0.47 **	1		
9. SF-12 (MCS)	0.16 **	0.18 **	−0.02	0.44 **	−0.67 **	−0.65 **	−0.54 **	0.87 **	1	
10. SF-12 (PCS)	0.04	0.09	0.00	0.29 **	−0.38 **	−0.33 **	−0.34 **	0.91 **	0.61 **	1

Note. HLC = health locus of control; HADS = Hospital Anxiety and Depression Scale; MCS = mental component summary; PCS = physical component summary; SF-12 = Short Form-12 Health Survey; * *p* < 0.05; ** *p* < 0.01.

**Table 3 jpm-13-00873-t003:** Longitudinal correlation analysis of all variables.

	Follow-Up (*n* = 323)									
Baseline	1	2	3	4	5	6	7	8	9	10
1. Other people HLC	0.47 **	0.24 **	0.22 **	0.18 **	−0.17 **	−0.13 **	−0.22 **	−0.06	0.05	−0.14 *
2. Internal HLC	0.22 **	0.44 **	−0.10	0.10	−0.14 *	−0.09	−0.16 **	0.13 *	0.16 **	0.08
3. Chance HLC	0.19 **	−0.01	0.43 **	0.04	−0.03	0.01	−0.06	−0.09	−0.08	−0.08
4. Positivity	0.23 **	0.17 **	0.02	0.54 **	−0.39 **	−0.35 **	−0.36 **	0.33 **	0.36 **	0.26 **
5. HADS	−0.24 **	−0.16 **	−0.04	−0.41 **	0.60 **	0.58 **	0.50 **	−0.35 **	−0.44 **	−0.22 **
6. Anxiety	−0.14 *	−0.13 *	0.00	−0.32 **	0.55 **	0.58 **	0.39 **	−0.31 **	−0.41 **	−0.16 **
7. Depression	−0.29 **	−0.19 **	−0.10	−0.43 **	0.51 **	0.41 **	0.54 **	−0.32 **	−0.37 **	−0.23 **
8. SF-12	0.12 *	0.06	−0.07	0.33 **	−0.45 **	−0.39 **	−0.39 **	−0.60 **	0.53 **	0.54 **
9. SF-12 (MCS)	0.18 **	0.09	−0.02	0.36 **	−0.51 **	−0.47 **	−0.44 **	0.49 **	0.55 **	0.36 **
10. SF-12 (PCS)	0.04	0.00	−0.11	0.23 **	−0.29 **	−0.22 **	−0.27 **	0.54 **	0.39 **	0.57 **

Note. HLC = health locus of control; HADS = Hospital Anxiety and Depression Scale; MCS = mental component summary; PCS = physical component summary; SF-12 = Short Form-12 Health Survey; * *p* < 0.05; ** *p* < 0.01.

## Data Availability

The data presented in this study are available on request from the corresponding author.

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
