# Peer review of "Positivity and Health Locus of Control: Key Variables to Intervene on Well-Being of Cardiovascular Disease Patients"

_jpm, 2023, doi:10.3390/jpm13050873_

Round 1

Reviewer 1 Report

1.The number of patients in baseline was 593 but in follow-up was 323, the authors should describe the number of patients in each Table, or should unify the number before all statistical analyses. Maybe the authors should re-analyze the data and rewrite all the Tables and Results based on only one number of patients.

2. Some terminologies or abbreviations, such as HRQoL, HRQOL, HQOL were not consistent.

3. In Figure 1, personality variables mean what? Positivity and HLC? Disposition variables mean what? Psychological wellbeing means what? They should be noted below the Figure 1.

4. While describing the results, the coefficients should be included in the section or paragraphs to help readers quickly understand according to the Figures. Such as below questions 5-7.

5.line 317-318, That “At follow-up no significant relationship were found between the independent and dependent variables.” was not consistent with the Figure 2, such as Anxiety had an effect of -.32 on Depression. What did you mean for independent and dependent variables?

6. line 323-325, “Crosssectionally, at Follow-up anxiety was positively related to depression and both anxiety and depression were negatively related to the global HRQoL.” Please added coefficients based on Figure 2.

7. line 330-332, “Finally, the educational level was positively related to positivity at baseline and with lower levels of depression at Follow-up.” Please added coefficients based on Figure 2.

8. In the Abstract, “The results revealed that both health locus of control and positivity were significantly correlated 19 with psychological well-being and health locus of control.” was confusing.

Author Response

The authors would like to thank you for the very helpful comments and time reviewing our manuscript. We have made several changes to our previous manuscript following your suggestion and comments and we tried to be clear and concise when addressing and respond your concerns. If you have any further suggestion, we will, of course, be pleased to include them. Please, find the responses (in blue) to each comment below.

Point 1. The number of patients in baseline was 593 but in follow-up was 323, the authors should describe the number of patients in each Table or should unify the number before all statistical analyses. Maybe the authors should re-analyze the data and rewrite all the Tables and Results based on only one number of patients.

Response to Point 1: Sociodemographic data were collected at the start of the study (N = 593). All analyses performed at that time were conducted with 593 participants. The follow-up analysis and SEM analysis were performed with 323 patients (the participants that completed both study phases). As suggested, this information has now been included in the tables (2a, 2b and 3, pages 9 and 10) and in the legend of Figure 2 (Standardized model parameter estimates, page 10).

Point 2. Some terminologies or abbreviations, such as HRQoL, HRQOL, HQOL were not consistent.

Response to Point 2: We appreciate this observation; we have fully revised and checked the manuscript in order to be consistent with the abbreviations.

Point 3. In Figure 1, personality variables mean what? Positivity and HLC? Disposition variables mean what? Psychological wellbeing means what? They should be noted below the Figure 1.

Response to Point 3. Thank you for this question. The information requested has been added in Figure 1 legend (Page 5).

Figure 1. Proposed research hypothesis model. Sociodemographic variables: age, sex, economic and educational level, etc. Personality and dispositional variables: Positivity and Health Locus of Control. Psychological well-being: anxiety and depression; HRQoL: Health-Related Quality of Life. The model does not include H2 and H3 as they refer to correlations between variables rather than predictive relationships.”

Point 4. While describing the results, the coefficients should be included in the section or paragraphs to help readers quickly understand according to the Figures. Such as below questions 5-7.

Response to Point 4. We agree with your recommendation. This information has been included in the results section (pages 7, 8 and 11).

The sub-sections 3.1 and 3.2 now read as follows:

“3.1. Cross-sectional analysis

In the first correlation analysis, data were obtained on the associations between all variables (at Baseline and at Follow-up independently). The significant correlations between the study variables were as follows (Table 2): at Baseline the other people HLC correlated negatively with depression (r = -.22) as well as with psychological distress (r = -.17); the internal HLC correlated negatively anxiety (r = -.15) and depression (r = -.22) and with psychological distress (r = -.22) and positively with all the HRQoL (global, r = .16; MCS, r = .14; PCS, r = .15). Moreover, the internal HLC was positively associated with the other people HLC factor (r = .26) and positivity (r = .13); the chance HLC correlated negatively with depression only (r = -.10). Finally, positivity correlated positively with the internal and other people HLC (r = .13, r = .20, respectively) and all HRQoL factors (global, r = .47; MCS, r = .51; PCS, r = .34), and negatively with psychological distress (r = -.55), depression (r = -.44), and anxiety (r = -.55). At Follow-up, the only differences were that the internal HLC did not positively correlate with the PCS HRQoL. Other people HLC correlated negatively with anxiety (r = -.19) and positively with MCS HRQoL (r = .16). Thus, most of the correlation results remained stable over time.

3.2. Longitudinal analysis

            The second correlation analysis was conducted to identify any statistically significant relationships between the evaluated variables at Baseline and at Follow-up (Table 3). Regarding the independent variables, the three HLC factors at Baseline significantly correlated with themselves at Follow-up (internal HLC, r = .47, other people HLC, r = .44) and chance HLC (r = .43). Only other people HLC was significantly associated with positivity at Follow up (r = .18). In contrast, baseline positivity correlated with the variables positivity (r = .54) other people HLC (r = .23) and internal HLC (r =.17) at Follow-up. On the other hand, with respect to the association between independent and dependent variables, several significant data were found. Baseline scores of the HLC other people correlated negatively with follow-up psychological distress (r = -.17), anxiety (r = -.13) depression (r = -.22) and PCS HRQoL (r = -.14); the internal HLC showed similar outcomes, but with the difference that this factor correlated positively with the global (r = .13) and MCS HRQoL (r = .16) and not with the PCS HRQoL and anxiety; the chance HLC did not correlate significantly with any dependent variable. Finally, baseline positivity correlated negatively with follow-up anxiety (r = -.35), depression (r = -.36) and psychological distress (r = -.39) and positively with all HRQoL factors (global, r = .33; MCS, r = .36; PCS, r = .26).

After performing correlation analyses, it was observed that the diagonal, i.e., the variables (dependent and independent) at baseline, correlated positively with themselves with a mostly large effect size (Cohen, 1988 [56]) at follow-up. Additionally, associations were found between the independent and dependent variables in these correlations (Table 3). Based on these findings, further comprehensive evaluations were conducted to examine the predictive potential of positivity and health locus of control (HLC) variables on psychological well-being, anxiety, depression, and HRQoL both cross-sectionally and longitudinally. To achieve this, a path analysis was performed. The model (Figure 1) demonstrated a strong fit to the data, with the following indices: χ2 (46, N = 514) = 41.636, p = 0.276; CMIN/DF = 1.125; CFI = .995; TLI = .991; AGFI = 0.957, GFI = 0.980 RMSEA = 0.020, 95% CI [.001, .046]). Figure 2 displays the standardized parameter estimates.

As it can be seen in Figure 2, at Baseline, positivity was negatively related to anxiety (r = -.42) and depression (r = -.45). The internal HLC was positively related to the other people HLC (r = .24),and negatively with chance HLC (r = -.20) but not with the dependent variables. Other people HLC was negatively related to depression (r = -.12). At follow-up no significant relationship were found between the independent (Positivity and HLC) and dependent variables (anxiety, depression and HRQoL). Longitudinally, positivity negatively predicted anxiety (r = -.15) and positively the global HRQoL (r =.16). The internal and other people HLC also predicted the global HRQoL, positively (r = .10) and negatively (r = -.16) respectively.

Besides these associations, other relationships between dependent variables both cross-sectionally and longitudinally were also found. For instance, anxiety and depression at baseline positively predicted anxiety (r = .48) and depression (r = .39) at Follow-up, respectively. Cross-sectionally, at Follow-up anxiety was positively related to depression (r = .40) and both anxiety and depression were negatively related to the global HRQoL (r = -.32 and, r = -.25, respectively).

Finally, the results of the path analysis showed some significant interactions between the sociodemographic variables age, sex and educational level that need to be acknowledged. Being female was associated with higher levels of anxiety (r = .21) and lower levels of internal HLC (r = -.13). The age was positively associated with other people (r = .19) and chance HLC (r = .15), and negatively with anxiety both at baseline (r = -.22) and at follow-up (r = -.14). Finally, the educational level was positively related to positivity at baseline (r = .26) and with lower levels of depression at Follow-up (r = -.10).”

Point 5. Line 317-318, That “At follow-up no significant relationship were found between the independent and dependent variables.” was not consistent with the Figure 2, such as Anxiety had an effect of -.32 on Depression. What did you mean for independent and dependent variables?

Response to Point 5. With this statement, we meant to report that no significant association was found between the independent (Positivity and HLC) and dependent variables (anxiety, depression and HRQoL). The effect of -.32 corresponds to an influence between dependent variables (anxiety and depression). We agree this sentence needs clarification. Therefore, following your question we have clarified in the 2.3. Statistical analysis section which were the dependent and independent variables. Now the readers will better understand the results section.   

2.3. Statistical analysis (page 7)

“The independent variables were “health locus of control (HLC)” and “positivity”, while the dependent variables were “psychological well-being,” “anxiety”, “depression” and HRQoL.”

Point 6. line 323-325, “Cross-sectionally, at Follow-up anxiety was positively related to depression and both anxiety and depression were negatively related to the global HRQoL.” Please added coefficients based on Figure 2.

Response to Point 6. We agree with your recommendation. This information has been included in the results section. Please see the Response to Point 4 to view the changes that have been made. As mentioned, this information has been reported in pages 7, 8 and 11 of the manuscript.

Point 7. line 330-332, “Finally, the educational level was positively related to positivity at baseline and with lower levels of depression at Follow-up.” Please added coefficients based on Figure 2.

Response to Point 7. We agree with your recommendation. This information has been included in the results section. Please see the Response to Point 4 to view the changes that have been made.

Point 8. In the Abstract, “The results revealed that both health locus of control and positivity were significantly correlated with psychological well-being and health related-quality of life.” was confusing.

Response to Point 8. Thank you for this comment. We have revised the abstract and we have corrected this information.  That sentence now reads as follows:

“The results revealed that both health locus of control and positivity were significantly correlated with psychological well-being and health related-quality of life.”

Reviewer 2 Report

ID: jpm-2350392

Title: Positivity and health locus of control: key variables to intervene on well-being of cardiovascular disease patients

Thank you for providing a chance to review this manuscript.

Comment: Rejection.

Detailed information:

Abstract

Line 14-16, Page 1: Please specify the specific time of this study.

Line 16-18, Page 1: Statistical analysis methods need to be more detailed and specific.

Line 18-19, Page 1: “The results revealed that both health locus of control and positivity were significantly correlated with psychological well-being and health locus of control” Positive or negative correlation? What is the r value? These are key results that need to be listed. In addition, the main statistical results of this study are also expected to be presented.

Introduction

Paragraph 1, Page 1: The background of cardiovascular disease needs to be further supplemented, such as definitions, incidence, etc.

Line 36-37, Page 1: “Psychosocial and personality factors can contribute to the risk of CVD or lead to a worse prognosis in patients with this disease”. Please supplement evidence.

Line 44-47, Page 2: The sudden emergence of positivity and health locus of control in the text is abrupt. What do they have to do with mental health? More description and context are needed to bridge the transition.

Line 64-65, Page 2: “This could explain the elevated prevalence of depression in CVD patients, which is higher than in the general population”. Is there specific data? It needs to be supported by references. Please note that all status descriptions require evidence.

Line 141-180, Page 4: “Aim and hypotheses” need to be simplified, the main assumptions of this research are supposed to present.

Overall: In general, the “Introduction” needs to be concise. 1.1 and 1.2 mainly describe the relationship between depression, anxiety, and positive psychology. The descriptions here are somewhat cumbersome, and some definitions can be shortened. For example, anxiety and depression can be described combined, and there is no need to split them into two descriptions. In addition, there is no relevant research basis and current situation for this study, please add relevant evidence.

Materials and Methods

Line 182-184, Page 4: The description of the study sample is too simplistic, and some important information needs to be added. What sampling method was used? When was the study conducted? What were the inclusion and exclusion criteria for the sample?

Line 185-190, Page 5: 1) Why is baseline and follow-up studies 9-month apart? 2) How is the sample size calculated? Is there a basis?

Line 200, Page 5: How to control bias in online surveys? How many questionnaires were given out? What is the recovery rate and effective rate respectively?

Line 212-213, Page 6: “Due to the low reliability of the doctor's factor in this study, it was excluded from the analysis”. Are the psychometric properties of the modified scale validated? Has the construct validity changed?

Line 214-218, Page 6: What is the factor structure of the P-scale? What is the score range? What does a higher score mean?

Line 243-254, Page 7: 1) Is the data checked? Does it conform to a normal distribution? 2) How are the results of statistical analysis evaluated? How to assess the outcome variables of correlation analysis and structural equation model?

Results

Line 255, Page 7: Supplement description of participants' sociological characteristics.

Line 257-271, Page 7: 1) Although correlation analysis results showed that HLC were statistically significant with other indicators, r value lower than 0.3 was generally considered to be meaningless. And in the correlation analysis of Positivity, r values were only moderately correlated. 2) What do HLC other people, HLC Internal and HLC Chance mean respectively? It's very confusing.

Line 273-286, Page 7-8: As previously suggested, the r values between the three HLC factors at baseline and other factors are too small to be convincing.

Figure 2: The diagram is too complex to discern, is there a more concise form?

Overall: For the description of the results, in addition to the text description, it is more important to list the statistical results, such as r and P values. In addition, the correlation analysis results are weak and insufficient.

Conclusions

Please simplify.

This study examined the effects of positivity and health locus of control to cardiovascular disease patients. First of all, the description of the process and method of this study is insufficient, and more details need to be added in the method. On the contrary, many descriptions in the introduction and discussion are redundant, and the author needs to reorganize the language to highlight the main points. Most importantly, the correlation analysis of this study is not in line with expectations. In addition, it is suggested that the author collects three longitudinal data and use a cross-hysteresis model to illustrate the causal relationship between the indicators. Overall, I'm not convinced by the results so far. It is a great pity that I made the decision to reject the article, and I look forward to further improving the quality of this article!

Thank you and my best,

Your reviewer

Moderate editing of English language

Author Response

The authors would like to thank you for the very helpful comments and time reviewing our manuscript. We have made several changes to our previous manuscript following your valuable suggestions and we tried to be clear and concise when addressing all your concerns. If you have any further suggestion, we will, of course, be pleased to include them. Please, find the responses (in blue) to each comment below.

Abstract

Point 1. Line 14-16, Page 1: Please specify the specific time of this study.

Response to Point 1: Thank you for this recommendation. For the current study, the Baseline assessment started in January 2017 and the Follow-up evaluation was conducted approximately 9 months later. As suggested, we have incorporated the information related to the specific time the study was conducted in the abstract as well as in the 2. Materials and Methods section.

Abstract (page 1):

“A total of 593 cardiac outpatients completed the Multidimensional Health Locus of Control questionnaire, the Positivity scale and the Hospital Anxiety and Depression Scale at baseline (January 2017) and 9-m later (Follow-up; n = 323).”

  1. Materials and Methods section (page 4)

2.1. Participants and procedure

“A longitudinal study was designed to test the hypotheses proposed. A battery of questionnaires was administered to the participants to assess HLC, positivity, psycho-logical well-being through the components of anxiety and depression, and HRQoL in two different times. Baseline assessment started in January 2017 and the Follow-up evaluation was conducted approximately 9 months later. At Baseline, the sample was composed of 593 patients, and at Follow-up of 323 CVD patients. Sociodemographic characteristics measured are shown in Table 1.”

Point 2. Line 16-18, Page 1: Statistical analysis methods need to be more detailed and specific.

Response to Point 2: We have specified in the Abstract section the analysis performed in the current study.

“Spearman rank correlation coefficient and a structural equation modeling approach were conducted to explore the relationships between health locus of control, positivity, psychological well-being, and health-related quality of life, both cross-sectionally and longitudinally.”

Point 3. Line 18-19, Page 1: “The results revealed that both health locus of control and positivity were significantly correlated with psychological well-being and health locus of control” Positive or negative correlation? What is the r value? These are key results that need to be listed. In addition, the main statistical results of this study are also expected to be presented.

Response to Point 3: Thank you for this comment. We have revised the abstract and we have corrected the information reported in this sentence. That sentence now reads as follows:

“The results revealed that both health locus of control and positivity were significantly correlated with psychological well-being and health related-quality of life.”

We have also indicated, as suggested, the direction of the association between variables:

“The results revealed that both health locus of control and positivity were positively correlated with psychological well-being and health-related quality of life.”

Introduction

Point 4. Paragraph 1, Page 1: The background of cardiovascular disease needs to be further supplemented, such as definitions, incidence, etc.

Response to Point 4. We agree with the reviewer that more conceptualization was needed in the first paragraph. Following your suggestion, we have amplified this information in Paragraph 1, which now reads as follows:

“Cardiovascular disease (CVD) is a broad term that encompasses various medical conditions affecting the heart and blood vessels [1]. These conditions include coronary artery disease, heart failure, arrhythmias, valvular heart disease, and peripheral artery disease, among others [1]. With its different typologies, CVD remains the leading cause of death worldwide [2]. To such an extent that according to the World Health Organization [3], in 2019, 17.9 million people died due to CVD, representing 32% of the global deaths. The enormous burden of this health condition [4] has led to the need for working at a preventive level, from health promotion to the implementation of interventions that reduce associated risk factors [5]. This aspect has become a real challenge for healthcare systems from all disciplines that address these types of diseases [6,7].”

Point 5. Line 36-37, Page 1: “Psychosocial and personality factors can contribute to the risk of CVD or lead to a worse prognosis in patients with this disease”. Please supplement evidence.

Response to Point 5: The introduction section has been reorganized and simplified following your suggestion. Therefore, this sentence is no longer in the manuscript. However, the idea reported in the sentence has been explained in the introduction section adding the references needed to support this statement along this section. Also, following your comment we have revised and incorporate more references and evidence in the introduction (pages 1-4).

Point 6. Line 44-47, Page 2: The sudden emergence of positivity and health locus of control in the text is abrupt. What do they have to do with mental health? More description and context are needed to bridge the transition.

Response to Point 6.  As previously mentioned, we have made several changes to the introduction section. Regarding this comment, we would like to note that we have introduced specifically the influence and need to evaluate personality dispositions to better understand and conceptualize cardiovascular disease (CVD) and their treatment from a psychological approach (page 2):

“… Therefore, from a biopsychosocial approach of health, interventions for CVD patients should also address their psychological well-being to mitigate the emotional consequences of the diagnosis [5,7,10]. To design effective integrative interventions for CVD patients, it is crucial to explore which psychological variables influence cardiovascular health (CVH) and well-being, as well as the role of personality dispositions on these variables. This approach could be beneficial both in the prevention and treatment of CVD, considering individual differences in psychological profiles [11,12]. For these reasons, incorporating psychosocial approach in cardiac rehabilitation could improve the clinical management of CVD and have a positive impact on patient outcomes [13].

First, we explained the relevance of psychological variables such as anxiety, depression, and health-related quality of life (HRQoL) in CVD (page 2):

“1.1. CVD and psychological well-being and quality of life 

Psychological well-being is a complex term to define, but it is typically associated with positive thoughts and emotions that individuals experience regarding their life satisfaction and overall sense of worth [14]. Therefore, it refers to optimal psychological functioning, which includes a combination of emotional aspects (e.g., happiness, experiencing positive emotions) as well as higher-level functions such as resilience, coping, and emotional regulation [14,15]. In contrast, psychological distress is composed of constructs such as anxiety, depression, anguish, or hostility [16].

Recent research has shown the association and relevance of psychological well-being or distress among patients with CVD [10,17,18]. It has been found that increased levels of well-being were related to improved CVH [10] and lower odds of mortality due to a cardiac event [5,19]. On the other hand, psychological distress, understood as depression and anxiety, has been found to be bidirectionally linked to CVD [8,20]. Meta-analyses and longitudinal studies have provided evidence that depression and anxiety are risk factors for CVD, with an association between them and an increased risk of developing heart diseases such as ischemic heart disease or coronary heart disease [21–23]. In addition, the diagnosis of CVD may exacerbate anxiety and depressive symptoms in these patients. Some studies have reported a high prevalence of depression among patients with coronary artery disease, with 34% of them experiencing moderate to severe depression [24]. This comorbidity could negatively impact the prognosis of the disease [24]. A meta-analysis performed by Gathright et al. [25] found that depression was a predictor of all-cause mortality in heart failure. Furthermore, patients with coronary heart disease or heart failure and depressive symptoms are more likely to have a lower quality of life, and a greater risk for recurrent cardiovascular events and mortality [22]. At the same time, studies addressing psychological distress and CVD showed some sex differences. Women with CVD reported greater and more severe symptomatology of anxiety and depression than men with CVD [26,27]. Additionally, it appears that psychological distress affects men and women with CVD differently in cardiac rehabilitation, resulting in higher dropout rates among women and lower rates of adherence to physical activity among men [28]. Consequently, it is important to consider these sex differences when designing and applying treatments for patients with CVD.

Cardiovascular patients can experience a significant impact on their lives, with consequences that encompass not only the psychological distress mentioned above (e.g., anxiety and depression) but also alterations at the physical level (e.g., reduced mobility, pain, and fatigue) [29]. These symptoms may become chronic and can negatively affect their quality of life [12]. Previous studies have shown the association between a CVD diagnosis and a lower quality of life, more specifically HRQoL [30,31]. Defined as a multidimensional construct, HRQoL includes the physical, mental, and social factors of an individual's health status, as well as their subjective perceptions of their own health and well-being [32]. At the same time, a lower HRQoL is associated with other risk factors in CVD patients, such as reduced adherence to medication and an increased risk of recurrent of cardiovascular events [33–35]. These findings have led to a wide body of research studying the role of HRQoL as an important variable in the context of chronic conditions, such as CVD, in order to better understand and intervene on its impact.

The impact of psychological distress on the quality of life of individuals with cardiovascular disease emphasizes the need to intervene in their emotional and psychological care [5–8,13]. Further research is necessary to identify key variables that promote their psychological well-being and provide protective effects against the disease.”

Then, we highlighted the importance of positivity and health locus of control (HLC) as personality dispositions that can impact the aforementioned psychological variables. This will help readers understand why these variables are key determinants in exploring the well-being of CVD patients (page 3):

“1.2. CVD and personality dispositions: Positivity and health locus of control

Related to well-being and quality of life, positivity is a psychological construct that provides insights into individuals' overall sense of wellness as it can be defined by factors such as self-esteem, satisfaction with life and optimism [38]. Factors that have been studied by their association with chronic conditions. For instance, self-esteem has been found to mediate the relationship between life satisfaction and lower depression in CVD patients [39]. Optimism on the other hand, seem to favor other psychological and cognitive mechanisms that promote better cardiovascular health (CVH) [10,17,40], reduce the risk of cardiovascular events [36] and facilitate the engaging of health-related behaviors [40]. In addition, several studies have found that positivity is a significant predictor of psychological variables related to psychological well-being and quality of life such as depression, anxiety [10,38,41] and also to the prognosis of the disease, reducing the rate of rehospitalization and mortality [41]. Therefore, encouraging individuals to adopt a more positive outlook on life, including their illness, would be necessary to help them feel more in control and self-efficacious in coping with their situation [17,36]. So, another much related and equally important variable would be the health locus of control.

Improving CVH prognosis involves making lifestyle changes [42], which can be influenced by various factors, including the personality disposition of health locus of control (HLC) [43]. HLC refers to people's beliefs about their ability to control their health, which can be either internal or external [43]. Internal HLC refers to individuals' perception of having control over their health, while external HLC refers to the belief that external factors, such as genetics, chance, or other people including family and physicians, have control over one's health [44]. According to the Health Locus of Control Theory, individuals’ health-related behaviors are associated with their perception of their ability to overcome health problems. Internal HLC has been found to be linked to engage in positive and protective health behaviors [45,46]. On the other hand, individuals with external HLC may have a lower sense of personal responsibility for one's own health, which may result in worse health outcomes and poorer disease prognosis [47].

The literature shows the significant role of psychological variables and personality traits in better understanding and conceptualizing the onset and consequences of CVD in a person's life. However, there is still a need to further investigate the specific mechanisms that underlie the association between these variables. Such research could provide valuable information for designing future psychological treatments and enhancing the effectiveness of cardiac rehabilitation programs, leading to improved psychological well-being and quality of life for CVD patients.”

Point 7. Line 64-65, Page 2: “This could explain the elevated prevalence of depression in CVD patients, which is higher than in the general population”. Is there specific data? It needs to be supported by references. Please note that all status descriptions require evidence.

Response to Point 7: In response to this comment, we have to report the changes made in the introduction section. Even if this sentence no longer appears in the manuscript, we have underlined the prevalence of depression and anxiety among CVD patients providing, as suggested, more evidence (please see Response to Point 6).

Point 8. Line 141-180, Page 4: “Aim and hypotheses” need to be simplified, the main assumptions of this research are supposed to present.

Response to Point 8: Thank you for this comment. We agree with this observation. Following your suggestion, we have considerably simplified the 1.3. aim and hypotheses section (page 4).

“The aim of this research is to explore the influence of HLC and positivity variables on psychological well-being, considering levels of anxiety and depression, and on HRQOL, in patients with CVD over time. For this purpose, the evaluations of these variables were carried out at two different times (Baseline and after approximately 9 months in a Follow-up). The hypotheses proposed for this study were as follows (Figure 1):

"H1. Sociodemographic variables are significantly associated with the dispositional variables, psychological well-being and the HRQoL.

H2. Positivity will positively correlate with psychological well-being as well as health related quality of life, both cross-sectionally (H2a) and longitudinally (H2b).

H3. Internal HLC will positively correlate with psychological well-being as well as health related quality of life, both cross-sectionally (H3a) and longitudinally (H3b).

H4. Positivity will predict the levels of psychological well-being and HRQoL variables in patients, both cross-sectionally (H4a) and longitudinally (H4b).

H5. Internal HLC will predict the levels of psychological well-being and HRQoL variables in patients, both cross-sectionally (H5a) and longitudinally (H5b)."

Point 9. Overall: In general, the “Introduction” needs to be concise. 1.1 and 1.2 mainly describe the relationship between depression, anxiety, and positive psychology. The descriptions here are somewhat cumbersome, and some definitions can be shortened. For example, anxiety and depression can be described combined, and there is no need to split them into two descriptions. In addition, there is no relevant research basis and current situation for this study, please add relevant evidence.

Response to Point 9: As per your feedback, we have revised the introduction section. We added more background information on cardiovascular disease that we felt was necessary. We also restructured subsections 1.1 and 1.2 to make them clearer and more concise. Furthermore, we included additional evidence and references throughout the introduction and streamlined subsection 1.3. We appreciate your comments as they have improved the overall understanding and quality of this section.

Materials and Methods

Point 10. Line 182-184, Page 4: The description of the study sample is too simplistic, and some important information needs to be added. What sampling method was used? When was the study conducted? What were the inclusion and exclusion criteria for the sample?

Response to Point 10: We agree that more information about study participants was needed. Following your suggestion, we have included the information requested in the participants and procedure subsection (2.1.). The paragraph now reads as follows (page 4):

“The study included 593 CVD patients (M = 64.75 SD = 9.07) recruited from the Cardiology Unit of the University Reina Sofía Hospital (Córdoba, Spain) who participated in the CORDIOPREV study [48,49] using a criterion sampling method. The inclusion and exclusion criteria of this study followed those of the primary study. Patients that had already suffered a first cardiac event and were diagnosed with an established coronary heart disease (e.g., unstable coronary disease, acute myocardial infarction, chronic coronary disease, unstable angina) were included. CVD patients that had experienced a clinical event in the last 6 months were excluded. The study sample characteristics had been previously published [48,49].”

Point 11. Line 185-190, Page 5: 1) Why is baseline and follow-up studies 9-month apart? 2) How is the sample size calculated? Is there a basis?

Response to Point 11: As discussed in section 2.1 Participants and Procedure (page 4), the participants in this study were also involved in a primary study (CORDIOPREV). To avoid causing additional burden to the patients, we deemed it appropriate to coincide their medical follow-up from CORDIOPREV [48,49] with the psychological assessment conducted in this study. This was the main reason of the 9-months delay between measurements.

    According the sample size, we did not perform a calculation as the participation on this study was based on the availability of participants of CORDIOPREV study. As mentioned in Response to Point 10. The original study where it can be seen the characteristics of the study sample are the references 48, 49 of the manuscript:

  1. Delgado-Lista, J.; Perez-Martinez, P.; Garcia-Rios, A.; Alcala-Diaz, J.F.; Perez-Caballero, A.I.; Gomez-Delgado, F.; Fuentes, F.; Quintana-Navarro, G.; Lopez-Segura, F.; Ortiz-Morales, A.M.; et al. CORonary Diet Intervention with Olive Oil and Cardiovascular PREVention Study (the CORDIOPREV Study): Rationale, Methods, and Baseline Characteristics: A Clinical Trial Comparing the Efficacy of a Mediterranean Diet Rich in Olive Oil versus a Low-Fat Diet on Cardiovascular Disease in Coronary Patients. American Heart Journal 2016, 177, 42–50, doi:10.1016/j.ahj.2016.04.011.
  2. Delgado-Lista, J.; Alcala-Diaz, J.F.; Torres-Peña, J.D.; Quintana-Navarro, G.M.; Fuentes, F.; Garcia-Rios, A.; Ortiz-Morales, A.M.; Gonzalez-Requero, A.I.; Perez-Caballero, A.I.; Yubero-Serrano, E.M.; et al. Long-Term Secondary Prevention of Cardiovascular Disease with a Mediterranean Diet and a Low-Fat Diet (CORDIOPREV): A Randomised Controlled Trial. Lancet 2022, 399, 1876–1885, doi:10.1016/S0140-6736(22)00122-2.

Point 12. Line 200, Page 5: How to control bias in online surveys? How many questionnaires were given out? What is the recovery rate and effective rate respectively?

Response to Point 12a: While we acknowledge the potential risk of bias associated with online surveys, our data collection was performed in situ. This approach allowed the researchers to respond and to assist with any inconvenience that arose during the evaluation. Additionally, we recognize that self-report questionnaires have their own inherent risk of bias, which we have acknowledged and included in the Limitations sub-section (page 13).

4.2. Limitations and future research

“There are some limitations in this study that need to be acknowledged and considered for future research. Firstly, the use of self-reporting questionnaires, even with validated instruments and guaranteed confidentiality and anonymity, can introduce bias into the data by relying solely on subjective reporting. Future studies may incorporate a multi-method assessment in order to obtain a more accurate information and reduce social desirability when collecting data (e.g., including external validation, honesty scales, etc.)…..”

Response to Point 12b: As outlined in the manuscript, participants were asked to complete a battery of questionnaires at two time points as part of the study evaluation. The battery included all instruments reported in the sub-Section 2.2 (page 6), starting with a sociodemographic ad hoc questionnaire, followed by the Multidimensional Health Locus of Control Scale, Positivity Scale, Hospital Anxiety and Depression Scale, and The Short Form-12 Health Survey. Therefore, at baseline, there were five instruments in the battery, and at follow-up, there were four (excluding the sociodemographic information already obtained).

Response to Point 12c: We conducted a longitudinal study to explore the role of certain personality variables on the psychological well-being and quality of life in CVD patients. As such, we did not perform any specific intervention, nor did we assess or have access to data related to patient recovery or effectiveness rates. However, the clinical implications of this study are at least significant to be considered. The results provide evidence for the relevance of these variables to HRQoL and psychological well-being. This, in turn, motivates the need to design future psychological interventions that can effectively analyze their effectiveness.

Point 13. Line 212-213, Page 6: “Due to the low reliability of the doctor's factor in this study, it was excluded from the analysis”. Are the psychometric properties of the modified scale validated? Has the construct validity changed?

Response to Point 13: Thank you for this question. To check the psychometric properties of the Multidimensional Health Locus of Control Scale after eliminating the doctor's factor in the current study, a confirmatory factor analysis was conducted at each measurement moment with results Indicating a good model fit of the measure.

Accordingly, the following sentence has been Included at the end of the description of the measure, 2.2. Instruments (page 6):

"However, after deleting this subscale, Confirmatory Factor Analyses indicated a good model fit of the scale at both measurement moments. The fit indices at T1 were: χ2(24) = 84.86, p < .001; RMSEA (90% CI) = .07 (.05, .08); CFI = .96; TLI = .95; GFI = .97; and AGFI = .94. At T2, the fit indices were: χ2(21) = 49.59, p = .002; RMSEA (90% CI) = .06 (.04, .08); CFI = .97; TLI = .96; GFI = .97; and AGFI = .93."

Point 14. Line 214-218, Page 6: What is the factor structure of the P-scale? What is the score range? What does a higher score mean?

Response to Point 14. Thank for outlining this aspect. We have incorporated this information in the 2.2. Instruments section (page 6). The description of this scale now reads as follows:

Positivity Scale (P-scale [37]). This scale evaluates the personal tendency to interpret life and vital experiences from a positive point of view. It is a unidimensional scale that contains eight items (e.g., “I have great faith in the future”) ranging from 1 (strongly disagree) to 5 (strongly agree). A higher score indicates greater positivity. Previous studies have reported adequate internal consistency of the measure in different countries, including Spain, with a Cronbach’s alpha of .81 [37]

Point 15. Line 243-254, Page 7: 1) Is the data checked? Does it conform to a normal distribution? 2) How are the results of statistical analysis evaluated? How to assess the outcome variables of correlation analysis and structural equation model?

Response to Point 15.  Thank you very much for this question. Following your comment, we have analyzed normality of each variable, found that they do not follow a normal distribution. Therefore, we followed the procedure used in previous literature when the assumption of normality is violated and conducted nonparametric tests (Nahm, 2016). Consequently, we have performed Spearman rank correlation coefficient analysis instead of Pearson and changed all the results in the manuscript both in text (pages 7 and 8) and in tables (2a and 2b, page 9)

The changes made on this aspect have been informed in the abstract section (page 1):

“Spearman rank correlation coefficient and a structural equation modeling approach were conducted to explore the relationships between health locus of control, positivity, psychological well-being, and health-related quality of life, both cross-sectionally and longitudinally.”

2.3 Statistical analysis section (page 7):

“Descriptive statistics were conducted to know the frequencies of the socio-demographic characteristics. Subsequently, Kolmogorov-Smirnov test was conducted to determine whether the data were normally distributed. The results conclude that the assumption of normality was violated in all evaluated scales. Therefore, we performed Spearman's rank correlation analyses to measure the association between the variables cross-sectionally and longitudinally

Regarding the second and third questions, we have included this information in the manuscript in the 2.3 Statistical analysis section (page 7):

“Then, we conducted a path analysis using the structural equation modelling (SEM) approach to further explore the explanatory capacity of the HLC and positivity on psychological well-being, anxiety and depression factors and Health-related quality of life. The model's adequacy was evaluated by means of several metrics, including the chi-squared statistic (χ2), the comparative fit index (CFI), the Tucker-Lewis index (TLI), the root mean square error of approximation (RMSEA), and the standardized root mean square residual (SRMR). For model evaluation, we followed Schermelleh-Engel et al. [55] recommendations. According to these authors, an acceptable model fit is indicated by a χ2/df value that is equal to or less than 3, as well as CFI and NNFI values that are equal to or greater than 0.95, and RMSEA values that are less than 0.08, accompanied by a Confidence Interval (CI) that is in close proximity to RMSEA. The independent variables were “health locus of control (HLC)” and “positivity”, while the dependent variables were “psychological well-being,” “anxiety”, “depression” and HRQoL. Descriptive analysis and Pearson’ correlations were performed using the statistic software SPSS (v.28) and to estimate the path coefficients we used the software package AMOS (v.13). To interpret correlation results, we considered Cohen's (1988) [56] suggestions, where a correlation coefficient of 0.1 to 0.3 was considered small, 0.3 to 0.5 was moderate, and greater than 0.5 was large. For both analyses we set the significance level at p < .05.”

References:

Nahm, F. S. (2016). Nonparametric statistical tests for the continuous data: the basic concept and the practical use. Korean Journal of Anesthesiology, 69(1), 8. https://doi.org/10.4097/kjae.2016.69.1.8

Results

Point 16. Line 255, Page 7: Supplement description of participants' sociological characteristics.

Response to Point 16. Following your suggestion, we have added a paragraph of description of participants' sociological characteristics at the beginning of the results section (page 8).

This paragraph reads as follows:

“Sociodemographic features of the participants are presented in Table 1. The study sample were majority composed by men [85.7%]. Regarding the laboral status, the most of the participants were retired [67%], followed by having a full-time job [20.1%], houseworkers [5.1%], unemployed [4.9%] and part-time workers [2.9%]. Regarding to their marital status, most patients had a partner [88.9%]. Finally, the maximum number of respondents have received middle educational qualification [56%] followed by high educational level [38.1%], low [3%], very high [2.7%] and very low [0.2%].”

Point 17. Line 257-271, Page 7: 1) Although correlation analysis results showed that HLC were statistically significant with other indicators, r value lower than 0.3 was generally considered to be meaningless. And in the correlation analysis of Positivity, r values were only moderately correlated. 2) What do HLC other people, HLC Internal and HLC Chance mean respectively? It's very confusing.

Response to Point 17a. Thank you for your comment. We appreciate your concern regarding the effect sizes of the r values. We acknowledge that following Cohen's (1998) effect size cut-off some of the r values reported in the result section are weak. However, in table 2(a) it can be seen that the r values correlating positivity and the other dependent variables (psychological distress (-.55), anxiety (-.44) depression (-.55) and HRQoL (global .46, MCS = .51 and PCS .34) are in majority moderate to large. At follow up it can be seen also moderate effects.

It is important to consider that recent studies have criticized the exigence of Cohen’s effect size classification. Gignac and Szodorai (2016) in their study reported that from a Meta-analytic search, less than the 3% of correlations in the literature were found to be as large as r = .50. It is suggested from different authors that power analysis should be redefine to 0.10 (small), 0.20 (moderate), and 0.30 (large). In addition, it is highly relevant to consider not only statistical significance but also clinical significance when interpreting study results. Therefore, the context of our study should be taken into account to determine the relevance of the findings.  For that it is crucial to differentiate and report both statistically significant results and the clinically relevant findings. Clinically significant findings can be described as those that result in improvements to the quality of life of individuals and may enhance medical care across physical, psychological, and social health domains (Sharma, 2021). In our study, we aimed to analyze the role of psychological variables in enhancing the psychological well-being and quality of life of CVD patients in order to better design cardiac interventions. Therefore, even if the results are statistically significant but have a “small” effect size, we consider them to be clinically relevant, as they can contribute to the improvement of medical care and the quality of life of patients.

We have included this information in the limitation section (page 13):

“There are some limitations in this study that need to be acknowledged and considered for future research. Firstly, the use of self-reporting questionnaires, even with validated instruments and guaranteed confidentiality and anonymity, can introduce bias into the data by relying solely on subjective reporting. Future studies may incorporate a multi-method assessment in order to obtain a more accurate information and reduce social desirability when collecting data (e.g., including external validation, honesty scales, etc.). In addition, reported correlations effect sizes could be considered as weak and or moderate according to Cohen (1988) [56]. However, recent literature criticizes the use of Cohen’s cut-off and proposes a more flexible classification [77], which would give our results greater power and validity. In any case, given the potential relevance of the results for therapeutic interventions, future studies with larger samples would be needed to detect stronger relationships between these variables. In line with this, to obtain a better understanding of the trajectory of outcomes and increase the statistical power, future studies may benefit from including more follow-up evaluations. Regarding to the last question, we appreciate this comment and we have incorporated more specific description of those constructs in the introduction section.”

Response to Point 17b. According to the HLC factors, we have included more information about them in the introduction and instruments sub-section to facilitate the comprehension of those concepts:

  1. Introduction (page 3):

“Improving CVH prognosis involves making lifestyle changes [41], which can be influenced by various factors, including the personality disposition of health locus of control (HLC) [42]. HLC refers to people's beliefs about their ability to control their health, which can be either internal or external [42]. Internal HLC refers to individuals' perception of having control over their health, while external HLC refers to the belief that external factors, such as genetics, chance, or other people including family and physicians, have control over one's health [43].”

2.2. Instruments (page 6):

"The MHLC-S evaluates the locus of control for health.  It is composed of four factors according to whom the control is assigned, which in this study are termed: internal HLC; chance HLC; doctors HLC; and other people HLC."

References:

Gignac, G. E., & Szodorai, E. T. (2016). Effect size guidelines for individual differences researchers. Personality and individual differences102, 74-78.

Sharma H. (2021). Statistical significance or clinical significance? A researcher's dilemma for appropriate interpretation of research results. Saudi journal of anaesthesia15(4), 431–434. https://doi.org/10.4103/sja.sja_158_21

Point 18. Line 273-286, Page 7-8: As previously suggested, the r values between the three HLC factors at baseline and other factors are too small to be convincing.

Response to Point 18. As discussed, we acknowledge the small effect. However, the primary objective of the study was not to investigate the association between HLC factors. In regard to HLC, we also recognize and discuss the minor impact on the dependent variables in the discussion section (page 11-13).

Point 19. Figure 2: The diagram is too complex to discern, is there a more concise form?

Response to Point 19. Following your suggestion, we have simplified the model. It is clearer now.

Point 20. Overall: For the description of the results, in addition to the text description, it is more important to list the statistical results, such as r and values. In addition, the correlation analysis results are weak and insufficient.

Response to Point 20. Following your suggestion as well as those of other reviewers, we have expanded the quantitative information presented in the results section of the text (pages 7, 8 and 11).

The sections 3.1 and 3.2 now read as follows:

“3.1. Cross-sectional analysis

In the first correlation analysis, data were obtained on the associations between all variables (at Baseline and at Follow-up independently). The significant correlations between the study variables were as follows (Table 2): at Baseline the other people HLC correlated negatively with depression (r = -.22) as well as with psychological distress (r = -.17); the internal HLC correlated negatively anxiety (r = -.15) and depression (r = -.22) and with psychological distress (r = -.22) and positively with all the HRQoL (global, r = .16; MCS, r = .14; PCS, r = .15). Moreover, the internal HLC was positively associated with the other people HLC factor (r = .26) and positivity (r = .13); the chance HLC correlated negatively with depression only (r = -.10). Finally, positivity correlated positively with the internal and other people HLC (r = .13, r = .20, respectively) and all HRQoL factors (global, r = .47; MCS, r = .51; PCS, r = .34), and negatively with psychological distress (r = -.55), depression (r = -.44), and anxiety (r = -.55). At Follow-up, the only differences were that the internal HLC did not positively correlate with the PCS HRQoL. Other people HLC correlated negatively with anxiety (r = -.19) and positively with MCS HRQoL (r = .16). Thus, most of the correlation results remained stable over time.

3.2. Longitudinal analysis

    The second correlation analysis was conducted to identify any statistically significant relationships between the evaluated variables at Baseline and at Follow-up (Table 3). Regarding the independent variables, the three HLC factors at Baseline significantly correlated with themselves at Follow-up (internal HLC, r = .47, other people HLC, r = .44) and chance HLC (r = .43). Only other people HLC was significantly associated with positivity at Follow up (r = .18). In contrast, baseline positivity correlated with the variables: positivity (r = .54) other people HLC (r = .23) and internal HLC (r =.17) at Follow-up. On the other hand, with respect to the association between independent and dependent variables, several significant data were found. Baseline scores of the HLC other people correlated negatively with follow-up psychological distress (r = -.17), anxiety (r = -.13) depression (r = -.22) and PCS HRQoL (r = -.14); the internal HLC showed similar outcomes, but with the difference that this factor correlated positively with the global (r = .13) and MCS HRQoL (r = .16) and not with the PCS HRQoL and anxiety; the chance HLC did not correlate significantly with any dependent variable. Finally, baseline positivity correlated negatively with follow-up anxiety (r = -.35), depression (r = -.36) and psychological distress (r = -.39) and positively with all HRQoL factors (global, r = .33; MCS, r = .36; PCS, r = .26).

After performing correlation analyses, it was observed that the diagonal, i.e., the variables (dependent and independent) at baseline, correlated positively with themselves with a mostly large effect size (Cohen, 1988 [56]) at follow-up. Additionally, associations were found between the independent and dependent variables in these correlations (Table 3). Based on these findings, further comprehensive evaluations were conducted to examine the predictive potential of positivity and health locus of control (HLC) variables on psychological well-being, anxiety, depression, and HRQoL both cross-sectionally and longitudinally. To achieve this, a path analysis was performed. The model (Figure 1) demonstrated a strong fit to the data, with the following indices: χ2 (46, N = 514) = 41.636, p = 0.276; CMIN/DF = 1.125; CFI = .995; TLI = .991; AGFI = 0.957, GFI = 0.980 RMSEA = 0.020, 95% CI [.001, .046]). Figure 2 displays the standardized parameter estimates.

As it can be seen in Figure 2, at Baseline, positivity was negatively related to anxiety (r = -.42) and depression (r = -.45). The internal HLC was positively related to the other people HLC (r = .24), and negatively with chance HLC (r = -.20) but not with the dependent variables. Other people HLC was negatively related to depression (r = -.12). At follow-up no significant relationship were found between the independent (Positivity and HLC) and dependent variables (anxiety, depression and HRQoL). Longitudinally, positivity negatively predicted anxiety (r = -.15) and positively the global HRQoL (r =.16). The internal and other people HLC also predicted the global HRQoL, positively (r = .10) and negatively (r = -.16) respectively.

Besides these associations, other relationships between dependent variables both cross-sectionally and longitudinally were also found. For instance, anxiety and depression at baseline positively predicted anxiety (r = .48) and depression (r = .39) at Follow-up, respectively. Cross-sectionally, at Follow-up anxiety was positively related to depression (r = .40) and both anxiety and depression were negatively related to the global HRQoL (r = -.32 and, r = -.25, respectively).

Finally, the results of the path analysis showed some significant interactions between the sociodemographic variables age, sex and educational level that need to be acknowledged. Being female was associated with higher levels of anxiety (r = .21) and lower levels of internal HLC (r = -.13). The age was positively associated with other people (r = .19) and chance HLC (r = .15), and negatively with anxiety both at baseline (r = -.22) and at follow-up (r = -.14). Finally, the educational level was positively related to positivity at baseline (r = .26) and with lower levels of depression at Follow-up (r = -.10).”

Regarding the comment “the correlation analysis results are weak and insufficient.” we would like to refer to our previous Response to Point 17, where we have already discussed the results and their effect size.

Conclusions

Point 21. Please simplify.

This study examined the effects of positivity and health locus of control to cardiovascular disease patients. First of all, the description of the process and method of this study is insufficient, and more details need to be added in the method. On the contrary, many descriptions in the introduction and discussion are redundant, and the author needs to reorganize the language to highlight the main points. Most importantly, the correlation analysis of this study is not in line with expectations. In addition, it is suggested that the author collects three longitudinal data and use a cross-hysteresis model to illustrate the causal relationship between the indicators. Overall, I'm not convinced by the results so far. It is a great pity that I made the decision to reject the article, and I look forward to further improving the quality of this article!

Response to Point 21. Thank you for your feedback on our study. We appreciate your comments and suggestions for improvement. We have revised the method section to provide more details and simplify the language in the introduction and discussion sections to highlight the main points. We will also take your suggestion and we will consider to amplify the time points evaluations we have incorporated it in the limitations and future research section (page 13):

“There are some limitations in this study that need to be acknowledged and considered for future research. Firstly, the use of self-reporting questionnaires, even with validated instruments and guaranteed confidentiality and anonymity, can introduce bias into the data by relying solely on subjective reporting. Future studies may incorporate a multi-method assessment in order to obtain a more accurate information and reduce social desirability when collecting data (e.g., including external validation, honesty scales, etc.). In addition, reported correlations effect sizes could be considered as weak and or moderate according to Cohen (1988) [56]. However, recent literature criticizes the use of Cohen’s cut-off and proposes a more flexible classification [77], which would give our results greater power and validity. In any case, given the potential relevance of the results for therapeutic interventions, future studies with larger samples would be needed to detect stronger relationships between these variables. In line with this, to obtain a better understanding of the trajectory of outcomes and increase the statistical power, future studies may benefit from including more follow-up evaluations…

We hope that these revisions had addressed your concerns and improve the quality of our article. Thank you again for your valuable feedback.

Reviewer 3 Report

Suggestion 1: 

Authors are invited to review the hypothesis model underlying their research. In section 1.3 five hypotheses are identified, and in Figure 1 only three are included to advance their contribution to baseline and to follow up. In technical term, it is recommended to separate the hypothesis from the prediction.

A hypothesis is a speculative proposition (or initial assumption) about how something works whose validity must be tested using experiments and the argument (If… then) is put forward in the present tense. In this article, the hypothesis highlighted seems to be as follows: high levels of positivity (understood as a combination of self-esteem, satisfaction with life and optimism) correspond to lower levels of anxiety, depression, and psychological discomfort, ultimately leading to greater well-being. In operational terms, H1 as illustrated in figure 1 is correct but in 1.3 is correct if that “will” disappears.

A prediction is an estimate of a future value and the argument (If… then) is put forward in positive or negative statements, advancing a fact, using a verb in future tense. Most notably, estimating the value of a dependent variable from the values of independent variables. Thus,

H2, H3, H4 and H5 should be formulated as predictions in 1.3 . Predictions are usually cast into quantitative form as probabilities with a certain degree of certainty and this is what appears in tables 2 and 3. However, there is a caveat in Table 3.

What is expected in the diagonal “baseline versus follow-up” variables is correlation 1.0 (the relationship of answers to the same scale in time A and B, in fact this diagonal appears as an assumption in Table 2a and b).  It is the test-retest practice that support predictive validity and reliability.

Suggestion 2

In Table 3 the highest empirical correlation is 0,64 (that is, 41,6% of variance overlaping among the answers obtained on the Hospital Anxiety and Depression Scale) and the lowest correlation is 0,43 (that is 18,5% of overlap in the Health Locus Control, subscale chance, that is an external control in patients: outcomes are due to the vagaries of fate and luck.

I miss a comment to the relevance of this diagonal in Table 3 because this is the territory of communalities supporting the path analysis illustrated in figure 2.

3 Suggestions for future research

In section 2, I miss a scale to measure sincerity, because patients are often not reliable in the responses they provide during the consultation or in the questionnaires.

I miss another scale that checks whether patients understand what they read and what they respond to. Effective communication is a two-way process, also with questionnaires.

Deliberate behaviors of distortion or exaggeration of symptoms aimed at obtaining an alleged reward is an issue that health care professionals may encounter in their clinical practice; however, research based reports are still uncommon during the 3rd decade of the 21st century. Conservative estimates suggest that 1 in 3 people with depression is imaginary invalid. Thus, in these correlation matrices and in this path analysis there are many patient-driven distortions.

Final suggestion

The law of parsimony prevails in science, and so eight hypothesis (H2a, b, H3a,b, H4a,b, H5a,b) tested only once with 593 CV patients is an excess. However, only one hypothesis and eight predictions derived provide a more acurate cognitive map.

Asking for the impossible

I suggest the authors to examine Hans Eysenck (1916-1997) approach to his hypothesis on extraversion as well on Neuroticism. It took him and collaborators years of experiments, scales and reports over the course of decades.

Author Response

The authors would like to thank you for your intellectual input, the very helpful comments and time reviewing our manuscript. We deeply appreciate your recommendations and suggestions. We have made several changes to our previous manuscript following your comments and we tried to be clear and concise when addressing and respond your concerns. If you have any further suggestion, we will, of course, be pleased to include them.   Please, find the responses (in blue) to each comment below.

Suggestion 1: 

Point 1a. Authors are invited to review the hypothesis model underlying their research. In section 1.3 five hypotheses are identified, and in Figure 1 only three are included to advance their contribution to baseline and to follow up. In technical term, it is recommended to separate the hypothesis from the prediction.

Response to Point 1a.  Thank you for your suggestion. Figure 1 shows the hypotheses of the predictive model. Hypotheses 2 and 3, which refer to the correlations between the variables studied, would indeed be missing. Therefore, it seems to us more clarifying not to include them in this model.

To clarify this, they have been included in the caption of the figure (page 5):

“Figure 1. Proposed research hypothesis model. Sociodemographic variables: age, sex, economic and educational level, etc. Personality and dispositional variables: Positivity and Health Locus of Control. Psychological well-being: anxiety and depression; HRQoL: Health-Related Quality of Life. The model does not include H2 and H3 as they refer to correlations between variables rather than predictive relationships.”

Point 1b. A hypothesis is a speculative proposition (or initial assumption) about how something works whose validity must be tested using experiments and the argument (If… then) is put forward in the present tense. In this article, the hypothesis highlighted seems to be as follows: high levels of positivity (understood as a combination of self-esteem, satisfaction with life and optimism) correspond to lower levels of anxiety, depression, and psychological discomfort, ultimately leading to greater well-being. In operational terms, H1 as illustrated in figure 1 is correct but in 1.3 is correct if that “will” disappear.

A prediction is an estimate of a future value and the argument (If… then) is put forward in positive or negative statements, advancing a fact, using a verb in future tense. Most notably, estimating the value of a dependent variable from the values of independent variables. Thus, H2, H3, H4 and H5 should be formulated as predictions in 1.3. Predictions are usually cast into quantitative form as probabilities with a certain degree of certainty and this is what appears in tables 2 and 3. However, there is a caveat in Table 3.

What is expected in the diagonal “baseline versus follow-up” variables are correlated (the relationship of answers to the same scale in time A and B, in fact this diagonal appears as an assumption in Table 2a and b).  It is the test-retest practice that support predictive validity and reliability.

Response to Point 1b. We really appreciate your comments and suggestions. Following your recommendation, we have considerably simplified the 1.3 aim and hypothesis section. This section now reads as follows (page 4):

“1.3. Aim and hypotheses

The aim of this research is to explore the influence of HLC and positivity variables on psychological well-being, considering levels of anxiety and depression, and HRQoL, in patients with CVD over time. For this purpose, the evaluations of these variables were carried out at two different times, thus being able to obtain results of the same variables in a first phase (Baseline), and after approximately 9 months in a second phase (Follow-up). The hypotheses proposed for this study were as follows (Figure 1):

H1. Sociodemographic variables are significantly associated with the dispositional variables, psychological well-being and the HRQoL.

H2. Positivity will positively correlate with psychological well-being as well as health related quality of life, both cross-sectionally (H2a) and longitudinally (H2b).

H3. Internal HLC will positively correlate with psychological well-being as well as health related quality of life, both cross-sectionally (H3a) and longitudinally (H3b).

H4. Positivity will predict the levels of psychological well-being and HRQoL variables in patients, both cross-sectionally (H4a) and longitudinally (H4b).

H5. Internal HLC will predict the levels of psychological well-being and HRQoL variables in patients, both cross-sectionally (H5a) and longitudinally (H5b)."

Regarding to the results, in Table 3, we have reported the correlation between the variables analyzed at baseline and the same variables at follow-up. It is important to note that we should not expect to see a correlation of 1.0 along the diagonal because we are evaluating the same variable at two different time points. This type of analysis is more akin to a test-retest measure. As you pointed out, this distinction is important to keep in mind when interpreting the results.

 Suggestion 2

Point 2. In Table 3 the highest empirical correlation is 0,64 (that is, 41,6% of variance overlapping among the answers obtained on the Hospital Anxiety and Depression Scale) and the lowest correlation is 0,43 (that is 18,5% of overlap in the Health Locus Control, subscale chance, that is an external control in patients: outcomes are due to the vagaries of fate and luck.

I miss a comment to the relevance of this diagonal in Table 3 because this is the territory of communalities supporting the path analysis illustrated in figure 2.

Response to Point 2. Thank you for this appreciation. As suggested, we have included a sentence highlighting the relevance of the Table 3 results in order to justify the path analysis performed after.

This paragraph now reads as follows (page 8):

“After performing correlation analyses, it was observed that the diagonal, i.e., the variables (dependent and independent) at baseline, correlated positively with themselves with a mostly large effect size (Cohen, 1988 [56]) at follow-up. Additionally, associations were found between the independent and dependent variables in these correlations (Table 3). Based on these findings, further comprehensive evaluations were conducted to examine the predictive potential of positivity and health locus of control (HLC) variables on psychological well-being, anxiety, depression, and HRQoL both cross-sectionally and longitudinally. To achieve this, a path analysis was performed. The model (Figure 1) demonstrated a strong fit to the data, with the following indices: χ2 (46, N = 514) = 41.636, p = 0.276; CMIN/DF = 1.125; CFI = .995; TLI = .991; AGFI = 0.957, GFI = 0.980 RMSEA = 0.020, 95% CI [.001, .046]). Figure 2 displays the standardized parameter estimates.”

 3 Suggestions for future research

Point 3. In section 2, I miss a scale to measure sincerity, because patients are often not reliable in the responses they provide during the consultation or in the questionnaires.

I miss another scale that checks whether patients understand what they read and what they respond to. Effective communication is a two-way process, also with questionnaires.

Deliberate behaviors of distortion or exaggeration of symptoms aimed at obtaining an alleged reward is an issue that health care professionals may encounter in their clinical practice; however, research-based reports are still uncommon during the 3rd decade of the 21st century. Conservative estimates suggest that 1 in 3 people with depression is imaginary invalid. Thus, in these correlation matrices and in this path analysis there are many patient-driven distortions.

Response to Point 3. Thank you for your comment. We are aware of the potential biases that may arise from self-reported measures. This type of research methodology is highly used in psychological research in all types of studies (e.g., longitudinal, randomized controlled trials). To mitigate this risk, we have selected validated instruments with high reliability (Cronbach’s alpha) and conducted the data collection in situ in a neutral environment to reduce the influence of external factors that may bias responses and also to address any difficulties that may arise during the evaluation. We have also ensured anonymity and confidentiality which may help to an honest and accuracy reporting. However, as you have pointed out, there are still potential biases, such as social desirability bias, that need to be acknowledged. Therefore, we have included this limitation in the discussion section for readers to consider. Additionally, we appreciate your suggestion and will incorporate it as future studies recommendation together with incorporate mixed-methods in order to gain reliability and validity of the results.

4.2. Limitations and future research (page 13):

“There are some limitations in this study that need to be acknowledged and considered for future research. Firstly, the use of self-reporting questionnaires, even with validated instruments and guaranteed confidentiality and anonymity, can introduce bias into the data by relying solely on subjective reporting. Future studies may incorporate a multi-method assessment in order to obtain a more accurate information and reduce social desirability when collecting data (e.g., including external validation, honesty scales, etc.). In addition, reported correlations effect sizes could be considered as weak and or moderate according to Cohen (1988) [56]. However, recent literature criticizes the use of Cohen’s cut-off and proposes a more flexible classification [77], which would give our results greater power and validity. In any case, given the potential relevance of the results for therapeutic interventions, future studies with larger samples would be needed to detect stronger relationships between these variables. In line with this, to obtain a better understanding of the trajectory of outcomes and increase the statistical power, future studies may benefit from including more follow-up evaluations.”

 Final suggestion

Point 4. The law of parsimony prevails in science, and so eight hypotheses (H2a, b, H3a, b, H4a, b, H5a, b) tested only once with 593 CV patients is an excess. However, only one hypothesis and eight predictions derived provide a more accurate cognitive map.

Response to Point 4. Thank you for this comment. Following your comment as well as the editor and other reviewers, in this section we have simplified the text considerably to facilitate the comprehension of the hypothesis proposed (see Response to Point 1b). Following the recommendations and comments we have maintained the format as hypothesis instead of predictions. However, we really appreciate your suggestion and we will consider the information for further studies.

 Asking for the impossible

Point 5. I suggest the authors to examine Hans Eysenck (1916-1997) approach to his hypothesis on extraversion as well on Neuroticism. It took him and collaborators years of experiments, scales and reports over the course of decades.

Response to Point 5. Thank you for this recommendation. Currently, we are following a line of research that explores the influence of psychosocial variables on the physical and psychological well-being of patients with chronic illness. We appreciate your suggestion; however, our choice of the positivity and locus of control variables is based on the recent and increasing literature that relates these variables to health, both physical and mental. Therefore, we consider it highly relevant to explore the role of these variables in the psychological well-being and quality of life of patients with cardiovascular disease. Nevertheless, extraversion and neuroticism can also be important variables to consider in future studies due to their relationship with key psychological variables such as positive and negative affect, social support, stress, optimism, health-related behaviors, among others. We are delighted to review the literature you have proposed for future studies addressing health psychology in chronic/cardiac patients.

Round 2

Reviewer 1 Report

1. The Figure 2 may appear a small error, that is, a path from which baseline sociodemographic to follow-up depression? Besides, in line 355, “Finally, the results of the path analysis showed some significant interactions between the sociodemographic variables age, sex and educational level that need to be acknowledged.” Did the authors mean that the sociodemographic variables had effects on dependent variables, such as anxiety, depression…? The term “Interactions” usually mean that two independent variables interact and then have an effect on a dependent variable. Please amend the term “interactions” in the sentence.

2.line 316, Were the authors sure that the path model (from baseline to follow-up) was executed with N=514? It was a different number of sample size, and it seemed impossible to have sample size higher than 323 since follow-up sample size was only 323.

3.The Table 3 was confusing, why the authors presented the right-top triangular values?

Author Response

Responses to Reviewer #1 comments

(Round 2)

Name of journal:

Journal of Personalized Medicine

Section:

Mechanisms of Diseases

Special Issue:

Current Challenges and Personalized Treatment in Cardiovascular Disease

Manuscript ID.:

jpm-2350392

Title:

Positivity and health locus of control: key variables to intervene on well-being of cardiovascular disease patients 

The authors would like to thank you for the very helpful comments and time reviewing our manuscript. We have made several corrections and changes to our previous manuscript following your suggestion and comments. We tried to be clear and concise when addressing and respond your concerns. If you have any further suggestion, we will, of course, be pleased to include them. Please, find the responses (in blue) to each comment below.

Point 1. The Figure 2 may appear a small error, that is, a path from which baseline sociodemographic to follow-up depression? Besides, in line 355, “Finally, the results of the path analysis showed some significant interactions between the sociodemographic variables age, sex and educational level that need to be acknowledged.” Did the authors mean that the sociodemographic variables had effects on dependent variables, such as anxiety, depression…? The term “Interactions” usually mean that two independent variables interact and then have an effect on a dependent variable. Please amend the term “interactions” in the sentence.

Response to Point 1: Thank you for this comment. Effectively there was a mistake in Figure 2, that has been already corrected. According to the sentence in Line 355, we meant that some association were also found between the sociodemographic variables and dependent ones. Thank you for this correction, we have changed the term of interactions for associations.

Figure 2. Standardized model parameter estimates (*p < 0.05; **p < 0.01; ***p < 0.001). HLC = Health Locus of Control; HRQoL = Health-Related Quality of Life. The figure only shows significant paths (N = 323).

This sentence now read as follows (Page 11):

“Finally, the results of the path analysis showed some significant associations between the sociodemographic variables age, sex and educational level that need to be acknowledged.”

Point 2. Line 316, Were the authors sure that the path model (from baseline to follow-up) was executed with N=514? It was a different number of sample size, and it seemed impossible to have sample size higher than 323 since follow-up sample size was only 323.

Response to Point 2: Thank you very much for this appreciation, we have revised this line in the manuscript and there was a mistake in the reported data. The indices of the model are now corrected and read as follows:

“The model (Figure 1) demonstrated a strong fit to the data, with the following indices: χ2 (37, N = 323) = 41.636, p = 0.276; CMIN/DF = 1.125; CFI = .995; TLI = .991; AGFI = 0.957, GFI = 0.980 RMSEA = 0.020, 95% CI [.001, .046]).”

Point 3. The Table 3 was confusing, why the authors presented the right-top triangular values?

Response to Point 3: Thank you for this question. We have included the right-top triangular values because in this table we have reported the correlation between the variables analyzed at baseline and the same variables at follow-up. It is important to note that we should not expect to see a correlation of 1.0 along the diagonal because we are evaluating the same variable at two different time points, which is similar to a test-retest measure. Therefore, all data reported in this table is valuable and relevant as it serves as the predecessor and justification for the path analysis. As recommended by another reviewer in the first round of revision, we added a sentence in the manuscript highlighting the importance of the Table 3 results to justify the path analysis performed afterwards.

This paragraph now reads as follows (page 8):

“After performing correlation analyses, it was observed that the diagonal, i.e., the variables (dependent and independent) at baseline, correlated positively with themselves with a mostly large effect size (Cohen, 1988 [56]) at follow-up. Additionally, associations were found between the independent and dependent variables in these correlations (Table 3).

Reviewer 2 Report

ID: jpm-2350392

Title: Positivity and health locus of control: key variables to intervene on well-being of cardiovascular disease patients

The authors have made some modifications to the manuscript according to the previous comments, but there are still some problems that have not been effectively modified. Same as the previous review comments, the correlation analysis was not ideal, leading me to doubt the reliability of the results of the structural equation model. Overall, I am still not convinced by the current explanation. Unfortunately, I once again chose to refuse the publication of this manuscript. Looking forward to seeing a more perfect presentation.

Comment: Rejection.

Detailed information:

Abstract

1) Please list the statistics of the main results, such as r-values, structural equation model results, etc.

2) The description of the abstract and the method are contradictory. Spearman rank correlation coefficient or Pearson's correlations?

Introduction

1) It is suggested that the author collect current research evidence, sort out the incidences and consequences of these phenomena, and express them in numerical form.

2) The author has made some changes to the introduction, but it is still a bit lengthy and needs to be further simplified.

Materials and Methods

1) What is "a criterion sampling method"? Random sampling, convenience sampling, snowball sampling, or some other sampling method?

2) Were the patients treated surgically? The psychological impact of surgical treatment on patients is also significant.

Results

1) The results of participants' sociological characteristics are listed as 3.1.

2) Although the authors believe that the correlation results of this study are sufficient, most of the current guidelines still support r at 0.3~0.5 is moderate, and generally expect above 0.5, such as the COSMIN guidelines. In addition, these indicators are considered clinically relevant, which needs to be supported by authoritative evidence. Therefore, I still think that the correlation is insufficient and the current interpretation does not convince me.

Conclusions

Please simplify.

 Thank you and my best,

Your reviewer

Moderate editing of English language.

Author Response

Response letter to Reviewer #2

(Round 2)

Name of journal:

Journal of Personalized Medicine

Section:

Mechanisms of Diseases

Special Issue:

Current Challenges and Personalized Treatment in Cardiovascular Disease

Manuscript ID.:

jpm-2350392

Title:

Positivity and health locus of control: key variables to intervene on well-being of cardiovascular disease patients 

Thank you for your time reviewing our manuscript and for your valuable feedback. We deeply appreciate your recommendations and suggestions. We have carefully considered your comments for further improving the manuscript and made several changes to our previous version. We believe that we have improved the quality and impact of our research with these corrections. We hope that you can see the potential in our research after this round of revision. If you have any further suggestion, we will, of course, be pleased to include them.  Please, find the responses (in blue) to each comment below.

Point 1. Abstract

  • Please list the statistics of the main results, such as r-values, structural equation model results, etc.

Response to Point 1.1: Following your suggestion, we have included the statistic information requested in the abstract section.

The results in this section now reads as follows:

Cross-sectional correlation analysis at baseline revealed that internal health locus of control and positivity were negatively associated with anxiety (rs = -.15 and -.44, ps < 0.01) and depression (rs = -.22 and -.55, ps < 0.01) and positively associated with health-related quality of life (rs = .16 and .46, ps < 0.01). Similar outcomes were found at follow up and in longitudinal correlations. According to the path analysis, positivity was found to be negatively associated to anxiety and depression level at baseline (β = -.42 and -.45, ps < 0.001). Longitudinally, positivity was negatively associated to depression (β =.15, p < 0.01) and together with internal health locus of control positively associated to health-related quality of life (β = .16 and .10, respectively, ps < 0.05).

  • The description of the abstract and the method are contradictory. Spearman rank correlation coefficient or Pearson's correlations?

Response to Point 1.2: Thank you for this observation. This information has been corrected in the method section. As discussed in the Round 1 of revision, we performed Spearman rank correlation coefficient given the non-assumption of normality.

The sentence in the 2.3. Statistical analysis now reads as follows (page 7):

“Descriptive analysis and Spearman’ correlations were performed using the statistic software SPSS (v.28) and to estimate the path coefficients we used the software package AMOS (v.13).”

Point 2. Introduction

  • It is suggested that the author collect current research evidence, sort out the incidences and consequences of these phenomena, and express them in numerical form.

Response to Point 2.1. In the first round of revision, we included more information related to the mortality rate of CVD. Following your suggestion, we have incorporated more numeric data regarding the prevalence and burden of CVD.

The first paragraph of the introduction section now reads as follows (page 1):

Cardiovascular disease (CVD) is a broad term that encompasses various medical conditions affecting the heart and blood vessels [1]. These conditions include coronary artery disease, heart failure, arrhythmias, valvular heart disease, and peripheral artery disease, among others [1]. CVD is highly prevalent worldwide, and this trend appears to be increasing. Recent data from the Global Burden of Cardiovascular Diseases and Risk Factors [2] shows that the total prevalence of CVD has nearly doubled in the last three decades, from 271 million in 1990 to 523 million in 2019. The increase in prevalence has been accompanied by a significant rise in disability-adjusted life years and years lived with disability from 17.1 million to 34.4 in the same period, as well as an alarming mortality rate among affected patients [2]. With its different typologies, CVD remains the leading cause of death worldwide [3]. To such an extent that according to the World Health Organization [4], in 2019, 17.9 million people died due to CVD, representing 32% of the global deaths. The enormous burden of this health condition [5] has led to the need for working at a preventive level, from health promotion to the implementation of interventions that reduce associated risk factors [6]. This aspect has become a real challenge for healthcare systems from all disciplines that address these types of diseases [7,8].”

2) The author has made some changes to the introduction, but it is still a bit lengthy and needs to be further simplified.

Response to Point 2.1. Following your suggestion, we have further simplified this section.

Point 3. Materials and Methods

  • What is "a criterion sampling method"? Random sampling, convenience sampling, snowball sampling, or some other sampling method?

Response to Point 3.1: We appreciate this observation. We have corrected the information reported regarding the sampling method. It was a convenience sampling as the sample of our study was recruited from patients participating in a primary study (CORDIOPREV)

This paragraph now reads as follows:

“The study included 593 CVD patients (M = 64.75 SD = 9.07) recruited from the Cardiology Unit of the University Reina Sofía Hospital (Córdoba, Spain) who participated in the CORDIOPREV study [48,49] using a convenience sampling method. The inclusion and exclusion criteria of this study followed those of the primary study. Patients that had already suffered a first cardiac event and were diagnosed with an established coronary heart disease (e.g., unstable coronary disease, acute myocardial infarction, chronic coronary disease, unstable angina) were included. CVD patients that had experienced a clinical event in the last 6 months were excluded. The study sample characteristics had been previously published [48,49].”

  • Were the patients treated surgically? The psychological impact of surgical treatment on patients is also significant.

Response to Point 3.2. Thank you for this question. As mentioned, participants on this study were also the study sample of a primary study (CORDIOPREV). Inclusion and exclusion criteria for CORDIOPREV study are reported in Delgado-Lista et al., (2016). The primary study's inclusion and exclusion criteria did not specifically mention surgical treatment as a requirement for participation. As a result, we did not have information on whether patients underwent surgical treatment. Nonetheless, we acknowledge that this could be a valuable topic for future studies. Despite not having specifically addressed this issue in our study, the aim of this research was to explore the role of personality dispositions in the quality of life and psychological well-being of patients with cardiovascular diseases in order obtain evidence to design effective psychological interventions for these patients. Therefore, future interventions that take into account the psychological variables addressed in our study could be incorporated into cardiac rehabilitation programs, benefiting patients who have undergone surgical interventions (e.g., heart transplant, coronary angioplasty, or coronary artery bypass graft) as well as those who have not.

The inclusion and exclusion criteria of the original study are presented below (Delgado-Lista et al., 2016):

“The patients were selected with acute coronary syndrome (unstable angina, acute myocardial infarction) and high-risk chronic coronary heart disease, according to the following criteria:

Acute myocardial infarction: The existence of at least two of the following three signs: angina-type chest pain (or anginal equivalents), typical ECG changes (appearance of new Q waves and/or changes in ST segments and/or T waves), and a rise in myocardial enzymes (CPK and/or CPK/MB> of twice the normal laboratory limits). The MB value criterion will prevail in case of discrepancies over the total CPK.

Unstable angina: Admission to hospital for angina-type chest pains lasting at least 15 minutes, both at rest and after exercise, which have increased in frequency and duration in recent days or weeks. The latest episode must have occurred at least 48 hours prior to admission and must be accompanied by at least one of the following electrocardiographic or analytical changes:

ST depression of at least 0.5 mm in 2 contiguous leads.

ST elevation of at least 1 mm in 2 contiguous leads.

T-wave inversion of at least 2 mm in 2 contiguous leads.

Positive troponin result.

Chronic high-risk ischemic heart disease: patients will be included who have been hospitalized for a coronary event and/or stable angina, at least once in the past 2 years and who have undergone diagnostic coronary angiography with evidence of severe coronary disease, which is defined as the existence of an epicardial vessel greater than 2.5 mm in diameter with stenosis of over 50%.”

Exclusion criteria:

Patients under 20 years of age or over 75 years old with a life expectancy of over 5 years.

Severe heart failure, NYHA functional class III or IV, with the exception of self-limited episodes of acute heart failure at the time of the acute ischemic event.

Severe left ventricular systolic dysfunction (with ejection fraction equal to or under 35%).

Patients with restricted capacity to follow the protocol: those unable to follow the prescribed diet for whatever reason, due to personal or family circumstances.

Risk factors which are severe or difficult to control: patients with hypertension and diabetes, where there is organ involvement that limits their survival, will be excluded (chronic renal failure with creatinine which is persistently >2.5 mg/dl) and disabling clinical manifestations of cerebral atherosclerosis.

Chronic diseases unrelated to coronary risk: severe psychiatric illnesses, chronic conditions requiring treatment that could modify the lipid metabolism (chronic renal failure, chronic liver disease, neoplasia under treatment, chronic obstructive pulmonary disease involving respiratory pulmonary failure with home oxygen therapy, endocrine diseases susceptible to decompensation and diseases of the digestive tract that involve episodes of diarrhea).

Participants in other studies: patients taking part in other studies will be excluded, at the time of selection or up to 30 days before the study begins.”

References:

Delgado-Lista, J., Perez-Martinez, P., Garcia-Rios, A., Alcala-Diaz, J. F., Perez-Caballero, A. I., Gomez-Delgado, F., Fuentes, F., Quintana-Navarro, G., Lopez-Segura, F., Ortiz-Morales, A. M., Delgado-Casado, N., Yubero-Serrano, E. M., Camargo, A., Marin, C., Rodriguez-Cantalejo, F., Gomez-Luna, P., Ordovas, J. M., Lopez-Miranda, J., & Perez-Jimenez, F. (2016). CORonary Diet Intervention with Olive oil and cardiovascular PREVention study (the CORDIOPREV study): Rationale, methods, and baseline characteristics: A clinical trial comparing the efficacy of a Mediterranean diet rich in olive oil versus a low-fat diet on cardiovascular disease in coronary patients. American heart journal, 177, 42–50. https://doi.org/10.1016/j.ahj.2016.04.011

Point 4. Results

  • The results of participants' sociological characteristics are listed as 3.1.

Response to Point 4.1. Thank you for this appreciation. The 3.1. subsection now represents the Sociodemographic characteristics of participants.

  • Although the authors believe that the correlation results of this study are sufficient, most of the current guidelines still support r at 0.3~0.5 is moderate, and generally expect above 0.5, such as the COSMIN guidelines. In addition, these indicators are considered clinically relevant, which needs to be supported by authoritative evidence. Therefore, I still think that the correlation is insufficient and the current interpretation does not convince me.

Response to Point 4.2. We appreciate your comment and concern in this aspect. we have carefully revised this comment in order to fulfill and address your concerns regarding the correlation results. Assuming that Spearman correlation coefficient can be sensitive to sample size, we appreciate your annotations and we have revised the COSMIN guidelines. From this literature, it can be noted that more than the absolute value of the correlation coefficient, it is very relevant the importance of having predefined hypotheses about expected correlations or changes (Angst, 2011). Therefore, following your suggestion and in lie with the COSMIN Guidelines we have incorporated a new hypothesis based on the changes that the correlation coefficients will experience over time (H4). As our study is longitudinal, this hypothesis states that we expect the direction of the correlation between dispositional and psychological variables to be consistent and similar in both assessments (baseline and follow-up). The results confirmed this hypothesis, which was discussed in the discussion section of the study.

New information has been added to the "1.2. CVD and Personality Dispositions: Positivity and Health Locus of Control" section to improve the clarity and readability of this new hypothesis.

1.2. CVD and personality dispositions: Positivity and health locus of control

“Personality dispositions are consistent patterns of thoughts, feelings, and behaviors that characterizes an individual and are relatively stable over time and across different contexts [36]. Some personality dispositions have been found to be related to well-being and quality of life, such as positivity and health locus of control (HLC). Positivity is a psychological construct that provides insights into individuals' overall sense of wellness as it can be defined by factors such as self-esteem, satisfaction with life and optimism [37].”

The hypothesis section now reads as follows (page 4):

“1.3. Aim and hypotheses

The aim of this research is to explore the influence of HLC and positivity variables on psychological well-being, considering levels of anxiety and depression, and HRQoL, in patients with CVD over time. For this purpose, the evaluations of these variables were carried out at two different times, thus being able to obtain results of the same variables in a first phase (Baseline), and after approximately 9 months in a second phase (Follow-up). The hypotheses proposed for this study were as follows (Figure 1):

H1. Sociodemographic variables are significantly associated with the dispositional variables, psychological well-being and the HRQoL.

H2. Patients with higher levels of positivity will have lower levels of anxiety and depression, and therefore greater psychological well-being as well as higher health re-lated quality of life, both cross-sectionally (H2a) and longitudinally (H2b).

H3. The patients with higher levels of internal HLC will have lower levels of anxiety and depression, and therefore greater psychological well-being and HRQoL, both cross-sectionally (H3a) and longitudinally (H3b).

H4. The internal HLC and Positivity, given the stability of personality dispositions, will maintain similar correlations with psychological well-being and HRQoL both at baseline and at follow-up.

H5. Positivity will predict the levels of psychological well-being and HRQoL variables (dependent variables) in patients, both cross-sectionally (H5a) and longitudinally (H5b).

H6. The internal HLC will predict the levels of psychological well-being and HRQoL variables (dependent variables) in patients, both cross-sectionally (H6a) and longitudi-nally (H6b).”

The inclusion of this hypothesis has been included in the Discussion section (page 11):

“The study's findings were also in line with H3b because internal HLC correlated positively with psychological well-being and HRQoL factors (except PCS) and negatively with depression, however, there was no significant relationship with anxiety. Furthermore, it should be noted that the factor of other people HLC showed similar associations than the internal HLC regarding the dependent variables, proving to be more related to the well-being of patients with CVD than anticipated. Finally, the results of the correlation analysis supported H4. The personality dispositions of internal HLC and positivity maintained similar correlations with anxiety, depression, and HRQoL at both assessment points, indicating consistent and stable associations between the analyzed variables.”

References:

Point 5. Conclusions

Please simplify.

Response to Point 5. We appreciate the feedback. Based on your comment, we have revised and simplified the conclusions section.

This section now reads as follows (page 14):

“This study highlights the important role that positivity and HLC play in psychological health outcomes for CVD patients. The findings suggest that promoting a positive orientation and internal HLC may lead to improved psychological well-being, reduced anxiety and depression levels, and enhanced HRQoL among these patients. In conclusion, the study results underline the importance of considering patients' psychological well-being in the context of cardiac rehabilitation, and suggests that interventions focused on a psychological approach may be beneficial for enhancing CVH and a better prognosis for these patients. Further studies are required in this direction in order to empirically investigate the effectiveness of incorporating this approach in cardiac care.”